



# Impact of millennial-scale oceanic variability on the Greenland ice sheet evolution throughout the Last Glacial Period

Ilaria Tabone[1,2], Alexander Robinson[1,2,3], Jorge Alvarez-Solas[1,2], and Marisa Montoya[1,2]

[1]Universidad Complutense de Madrid, 28040 Madrid, Spain
[2]Instituto de Geociencias, Consejo Superior de Investigaciones Cientificas-Universidad Complutense de Madrid, 28040 Madrid, Spain
[3]Now at Faculty of Geology and Geoenvironment, University of Athens, 15784 Athens, Greece

*Correspondence to:* Ilaria Tabone (itabone@ucm.es)

**Abstract.** Temperature reconstructions from Greenland ice sheet (GrIS) ice cores indicate the occurrence of more than twenty abrupt warmings during the Last Glacial Period (LGP) known as Dansgaard-Oeschger (D-O) events. Although their ultimate cause is still debated, evidence from both proxy data and modelling studies robustly links these to reorganisations of the Atlantic Meridional Overturning Circulation (AMOC). During the LGP, the GrIS expanded as far as the continental shelf break and was thus more directly exposed to oceanic changes than in the present. Therefore oceanic temperature fluctuations on millennial timescales could have had a non-negligible impact on the GrIS. Here we assess the effect of millennial-scale oceanic variability on the GrIS evolution from the last interglacial to the present day. To do so, we use a three-dimensional hybrid ice-sheet-shelf model forced by oceanic fluctuations derived from paleo records. We show that the GrIS evolution during the LGP could have been strongly influenced by oceanic changes on millennial timescales, leading to ocean-induced ice volume contributions more than 1.5 m SLE. Several regions across the GrIS could thus have been responsible for ice mass discharge during D-O events, opening the possibility of a non-negligible role of the GrIS in oceanic reorganisations throughout the LGP.

## 1 Introduction

Ice cores from the Greenland ice sheet (GrIS) indicate that the climate of Greenland experienced multiple abrupt temperature increases during the Last Glacial Period (LGP) known as Dansgaard-Oeschger (D-O) events (Dansgaard et al., 1993). D-Os have waiting times between consecutive events of around 1500 years and, with decreasing probability, 3000 and 4500 years (Alley et al., 2001) and are characterised by an initial abrupt warming of 10-16 °C (interstadial state) followed by a slow cooling which finally ends with an abrupt return to glacial background conditions (stadial state) (Johnsen et al., 1992; Kindler et al., 2014). These millennial-scale climate fluctuations were first observed in Greenland ice cores (Anklin et al., 1993; Grootes et al., 1993), but strong evidence is also found in numerous marine sediments records, especially in the Nordic Seas and North Atlantic (Bond et al., 1993; Rasmussen et al., 1996; Voelker et al., 1997), tropical and subtropical stalagmite proxy archives (Fleitmann et al., 2009; Wang et al., 2001), and in Antarctic Ice Sheet (AIS) ice cores (Barbante et al., 2006), suggesting a worldwide imprint of D-O events (Voelker, 2002).



The numerous and extensive evidence from paleo records (e.g. Shackleton et al. (2000); Böhm et al. (2015); Gottschalk et al. (2015); Henry et al. (2016)) and from results from a variety of models, ranging from Earth system Models of Intermediate Complexity (EMICs) to comprehensive General Circulation Models (GCMs) (e.g. Ganopolski and Rahmstorf (2001); Shaffer et al. (2004); Peltier and Vettoretti (2014); Vettoretti and Peltier (2016, 2018)) widely attribute the cause of D-O events to reorganisations of the Atlantic Mean Overturning Circulation (AMOC), which are likely modulated by variations in the oceanic convection over the Nordic seas and North Atlantic (Dokken and Jansen, 1999; Rasmussen and Thomsen, 2004; Dokken et al., 2013; Barker et al., 2015; Hoff et al., 2016; Rasmussen et al., 2016); a recent review on the available evidence was provided by Lynch-Stieglitz (2017). However, the final cause of these AMOC reorganisations is still unclear. Numerous modelling studies have explored the problem. Many works attribute AMOC instability either to freshwater discharge from the Northern Hemisphere (NH) ice sheets (Ganopolski and Rahmstorf, 2001; Vellinga and Wood, 2002; Menviel et al., 2014; Bagniewski et al., 2017) directly connected to changes in the strength (Skinner and Elderfield, 2007) and in the location (Sévellec and Fedorov, 2015) of deep convection. Other possible mechanisms link the origin of D-O events to sea-ice cover variability (Li et al., 2005, 2010), or to linked sea ice-ice shelf fluctuations (Petersen et al., 2013). Still others connect AMOC reorganisations to climatic perturbations in the atmosphere associated with changes in ice-sheet dynamics (Wunsch, 2006; Zhang et al., 2014a), to progressive $CO_2$ atmospheric variations (Zhang et al., 2017), to changes in atmospheric heat transport (Wang et al., 2015), or through combined changes in wind and atmospheric $CO_2$ amount driven by the Southern Ocean (Banderas et al., 2015). Recently, D-O events have been explained as the result of a non-linear internal salt oscillator in the Atlantic (Peltier and Vettoretti, 2014; Vettoretti and Peltier, 2016, 2018) without the need to invoke any external forcing.

Despite the evidence for millennial-scale climate variability during the LGP in Greenland, the specific role of the GrIS in this framework has not been investigated in depth. Whether the GrIS was an active or a passive player in this oscillatory system is unknown and its evolution between the last interglacial and the last glacial maximum (LGM) is still debated (Vasskog et al., 2016). Evidence of ice-rafted debris (IRD) deposition in marine sediment cores suggests that the GrIS could have been a non-negligible source of iceberg discharge during the LGP (Jonkers et al., 2010). Cores drilled in the Irminger Sea (Van Kreveld et al., 2000), Denmark Strait (Bond and Lotti, 1995) and close to Scoresby Sund (Stein et al., 1996) show evidence of iceberg transport sourced in East Greenland and the North-Northeast Greenland, among other sources (Andrews et al., 2017); also, sediments in Fram Strait show imprints of iceberg discharge from the Northern GrIS (Darby et al., 2002). Still, others link the IRD deposition found close to the east-southeast marine margin of the GrIS to local ice-sheet instability (Verplanck et al., 2009), and associate sediments found in the Labrador Sea to iceberg discharge coming from Baffin Bay during the Younger Dryas (Andrews et al., 2012). Also, other paleo reconstructions explicitly affirm that ocean temperature increase played a fundamental role in the GrIS retreat in the last deglaciation (Jennings et al., 2017). Recent proxy data collected by morain-derived marine shells suggest that the margin of the Northeast Greenland Ice Stream (NEGIS) region may have fluctuated throughout the LGP by more than 200 km (Larsen et al., 2018). All these examples suggest that the GrIS may have experienced substantial variability during the LGP and beyond.

From a modelling perspective, only a few studies have tackled the response of the GrIS to millennial-scale climate variability throughout the LGP. Charbit et al. (2007) simulated the evolution of NH ice sheets by forcing their ice-sheet model with a

glacial-interglacial climate anomaly scaled by a climatic index taken from the GRIP reconstruction, which retained millennial-scale temperature fluctuations. However, their model accounted only for grounded ice; ice shelves were not included and the effect of the ocean was thus not taken into account. Huybrechts (2002) also investigated the response of the GrIS to millennial-

scale climate variability during the LGP with an ice-sheet model. However, the GrIS extent was controlled by orbital-only variations of past eustatic sea-level and millennial-scale fluctuations, both in sea level and in the ocean temperature, were omitted. Marshall and Koutnik (2006) assessed the response of the NH ice sheets to climate variability at millennial timescales. They simulated the GrIS evolution using an ice-sheet model that included a calving parameterisation, and that was forced by millennial-scale temperature variations. Their results showed very little response of the GrIS to the imposed climate variability,

with a weak increase in iceberg flux only during interstadials. However, this study did not explicitly investigate the effect of the millennial-scale variability in the ocean either.

A recent study has demonstrated the important role of oceanic conditions in the evolution of the GrIS throughout the past two glacial cycles (Tabone et al., 2018). This study however focused on orbital oceanic variations. Thus, the effect of oceanic millennial-scale variations on the GrIS remains to be explored. Sea surface temperatures (SSTs) inferred from planktonic

foraminifera in North Atlantic sediment cores vary typically from 7-8.5 °C during interstadials to 3-3.5 °C during stadials (Rasmussen et al., 2016). Recent SST estimates from a stack of sediment cores of the North Atlantic suggest a broad range of interstadial-stadial temperatures fluctuating between a maximum of 15 °C and a minimum of 0 °C, but typical SSTs anomalies in each location are lower (2-6 °C) (Jensen et al., 2018). The oceanic temperature variation associated with D-O events may thus have an appreciable impact on the GrIS. Here we use a three-dimensional, thermomechanical, hybrid ice-sheet-shelf

model to investigate this issue. Stadial/interstadial temperature variations in the ocean are expressed in the model through a linear parameterisation of the submarine melting rate at the grounding line and below the ice shelves. We characterise the millennial-scale variability in the ocean by performing two large ensembles of simulations, one forced by both orbital and millennial components in the oceanic temperature signal and the other forced by the solely orbital frequencies. In both ensembles, the atmospheric forcing only includes orbital changes, so comparison between the two large ensembles returns the

impact of millennial-driven oceanic fluctuations on the GrIS evolution. Moreover, we investigate the impact of the oceanic-induced perturbations at the marine margin onto the GrIS interior. This gives further insight into the potential role of the ocean in dynamic reorganisations of the ice sheet and enhanced ice discharge.

The paper is structured as follows: the ice-sheet-shelf model, the forcing methods and the methodology used to study the sensitivity of the GrIS to millennial fluctuations in the submarine melting are described in Section 2; in Section 3, we first

characterize the millennial-scale variability effect in the ocean on the GrIS evolution and we describe the impact of this oceanic oscillation on the GrIS transient dynamics. We then discuss results and caveats of the model, followed by the conclusions.



## 2 Tools and methods

### 2.1 GRISLI-UCM ice-sheet-shelf model

Here we use the three-dimensional, thermo-mechanical, hybrid ice-sheet-shelf model GRISLI-UCM (Alvarez-Solas et al., 2017; Tabone et al., 2018), developed from the GRISLI ice-sheet model (Ritz et al., 2001) which has been substantially modified through diverse parameterisations of ice-sheet boundary conditions (e.g. surface mass balance, basal sliding and basal
melt). The model solves deformational-flow-driven grounded areas via the shallow-ice approximation (SIA, (Hutter, 1983)) and floating shelves predominantly moving under plug flow via the shallow-shelf approximation (SSA, (Morland et al., 1987)). Transitional areas where these two flow regimes coexist are solved by adding the velocities of the SIA and SSA solutions (Winkelmann et al., 2011). The ice base of SIA-solved areas is supposed to be in a cold phase (below the pressure melting point) and does not admit sliding. On the contrary, transitional zones conditionally allow for basal sliding as long as the ice base
is temperate (at the pressure melting point) and the effective pressure at the bed is below a critical threshold. Once this condition is satisfied, the basal shear traction is calculated to be proportional to the ice-base velocity and to a coefficient depending on the basal water pressure and the bed roughness. Glacial isostatic adjustment (GIA) of the bedrock related to changes in the ice pressure is calculated through the Elastic Lithosphere Relaxing Asthenosphere model (Le Meur and Huybrechts, 1996). Grounding-line migration is expressed through a flotation condition between the temporally-variable sea level prescribed in the
model and the ice thickness calculated at the ice-ocean interface. In this study the position of the grounding line is diagnosed at sub-grid scale precision by interpolating the ice thickness over the grid cell including the grounding line (adapted from Gladstone et al. (2010)). The calculation returns the grounded percentage of the grid cell and avoids the simplistic assumption for which the grounding line lies on the last grounded coarse-grid point before floating ice.

The total ice mass balance is defined at any time as the difference between the mass balance at the ice surface (accumulation
minus ablation), basal melting at the ice base and ice calving. Surface ablation is calculated by the positive degree day (PDD) scheme (Reeh, 1989). This scheme is known to be overly simplistic for both ice sheet models (Robinson and Goelzer, 2014) and EMICs (Bauer and Ganopolski, 2017) in the paleo context, as it does not incorporate the effect of incoming solar radiation changes. Nevertheless, since here we focus on the sensitivity of the ice sheet to millennial-scale oceanic variations during the LGP, the choice of this scheme should be sufficient for our purposes. Surface precipitation is exponentially proportional to
atmospheric temperatures, which vary through an index approach (Banderas et al., 2018; Blasco et al., 2018; Tabone et al., 2018) (Section 2.2). Geothermal heat flux exchanged at the base of grounded ice is taken from Shapiro and Ritzwoller (2004). Basal melting at the grounding line and at the ice-shelf base is described in detail in Section 2.3. Calving is calculated following a two-rule criterion: the ice front must not exceed a critical thickness (H=200 m) and advected ice from upstream must fail to maintain the ice thickness above that imposed threshold (Peyaud et al., 2007; Colleoni et al., 2014).
The GRISLI-UCM model is applied here to the GrIS domain, with a spatial resolution of 20 km by 20 km. Since the goal of this work is to investigate the sensitivity of the GrIS to past millennial-scale variability in the ocean, we consider the atmosphere as modulated only by orbital changes for simplicity. This allows us to study the direct effects of the millennial-scale fluctuations on the GrIS evolution due to the ocean only, unperturbed by the millennial variability in the atmosphere too.



## 2.2 Atmospheric forcing

The atmospheric forcing includes the orbital-scale evolution of temperature and precipitation over the last glacial cycle, thus excluding millennial-scale atmospheric variability. As in Tabone et al. (2018), the atmospheric temperature forcing applied to the model follows an index-anomaly method in which the present-day temperature $T_{\text{atm}}^{\text{clim}}$ is modified by fluctuations of past temperature anomalies $\Delta T_{\text{atm}}(t)$:

$$T_{\text{atm}}(t) = T_{\text{atm}}^{\text{clim}} + \Delta T_{\text{atm}}(t) \tag{1}$$

where

$$\Delta T_{\text{atm}}(t) = (1 - \alpha(t))\,\Delta T_{\text{atm}}^{\text{orb}} \tag{2}$$

The present-day temperature climatology $T_{clim,atm}$ is obtained from the regional climate model MAR forced by the ERA-Interim reanalysis (Fettweis et al., 2013) for the period 1981-2010. The resulting atmospheric temperature $T_{\text{atm}}(t)$ is a 2D field which varies in time according to $\alpha(t)$, which is a spatially-uniform climatic index built through a composite series of various proxy-derived temperature anomalies from the Last Interglacial period to the present day (Tabone et al., 2018) (Fig. 1a). The index is then smoothed to remove the spectral components below the orbital frequencies (1/f < 18 ka) and is normalized between 0 and 1 ($\alpha = 1$ at present day (PD) and $\alpha = 0$ at the LGM) (Fig. 1b). Thus, by construction, $\Delta T_{\text{atm}}^{\text{orb}}$ is the glacial-interglacial atmospheric temperature anomaly, which is here taken from glacial minus present-day anomaly snapshots of the EMIC CLIMBER-3$\alpha$ model (Montoya and Levermann, 2008). This orbital forcing method is likewise applied to the precipitation field, which is parameterised through the ratio of glacial and present-day precipitation anomalies $\delta P_{\text{ann}}^{orb}$, as in Tabone et al. (2018):

$$P_{\text{ann}}(t) = P_{\text{ann}}^{\text{clim}} \cdot (\alpha(t) + (1 - \alpha(t)) \cdot \delta P_{\text{ann}}^{orb}) \tag{3}$$

## 2.3 Oceanic forcing

The methodology used to force the ocean is similar to that of Tabone et al. (2018), except that here we include the millennial-scale variability in the ocean which was omitted in the previous work. This approach is analogous to that of Blasco et al. (2018) for the Antarctic domain. Here, the basal melting rate at the grounding line is parameterised as:

$$B(t) = \kappa(T_{\text{ocn}}^{\text{clim}} - T_f) + \kappa\,\Delta T_{\text{ocn}}(t) \tag{4}$$

where

$$\Delta T_{\text{ocn}}(t) = (1 - \alpha(t))\,\Delta T_{\text{ocn}}^{\text{orb}} + \beta\,\Delta T_{\text{ocn}}^{\text{mil}} \tag{5}$$

The formulation is adapted from Beckmann and Goosse (2003) and permits the translation of changes in temperature into changes in basal melt through an ocean-ice heat-flux exchange scaling factor $\kappa$ ($m\ a^{-1}\ K^{-1}$). The first term of Eq. 4 is the





prescribed basal melting rate for the present day, where $T_f$ is the temperature at the base of the ice shelf, assumed to be at the freezing point, $T_{\text{ocn}}$ is the temperature of the ocean at the present day and $T_{\text{ocn}}^{\text{clim}}$ is the mean climatological temperature of the ocean. Since the present-day basal melt rate for the whole Greenland domain is largely unconstrained, we assume it to be spatially constant (hereafter referred to as $B_{\text{ref}} = \kappa (T_{\text{ocn}}^{\text{clim}} - T_f)$ ($m\ a^{-1}$)). The reference basal melt is then perturbed by its anomaly in time, which is here the sum of both orbital and millennial variability in the oceanic temperature. $\beta$ (Fig. 1c) is

the millennial index derived from the same normalized temperature anomaly signal as $\alpha$, by subtracting the orbital $\alpha$ index (Fig. 1b) from the unfiltered multi-proxy temperature signal (Fig. 1a) normalized between 0 and 1, and then filtered to remove periods below 1 ka to eliminate sub-millennial components of the signal. $\Delta T_{\text{ocn}}^{\text{orb}}$ ($K$) and $\Delta T_{\text{ocn}}^{\text{mil}}$ are the glacial-interglacial and interstadial-stadial oceanic temperature anomaly ($K$), respectively, both assumed to be in phase with the atmosphere.

As in Tabone et al. (2018) and Blasco et al. (2018), basal melt under the ice shelves is 10% of that near the grounding line, as
supported by recent estimates in Greenland (Rignot and Steffen, 2008; Münchow et al., 2014; Wilson et al., 2017). To prevent unconstrained accretion below ice shelves and at the grounding line, negative basal melt rate (freezing) is cut off to $0\ m\ a^{-1}$. Changes in sea level at orbital timescales are prescribed following Bintanja and Van de Wal (2008).

### 2.4 Perturbed basal melting parameters and LE description

Following the discussion above, we can rewrite the basal melting equation (Eq. 4) as:

$$B(t) = B_{\text{ref}} + \kappa \left( (1 - \alpha(t)) \Delta T_{\text{ocn}}^{\text{orb}} + \beta \Delta T_{\text{ocn}}^{\text{mil}} \right) \qquad (6)$$

It is therefore clear that the basal melting formulation depends on the choice of four parameter values: $B_{\text{ref}}$, $\kappa$, $\Delta T_{\text{ocn}}^{\text{orb}}$ and $\Delta T_{\text{ocn}}^{\text{mil}}$. Here, these are all considered as spatially uniform around Greenland for the sake of simplicity. To assess the GrIS response to millennial-scale variability in the ocean we could simply consider varying the value of $\kappa$, which is the sensitivity of the oceanic forcing (Tabone et al., 2018). However, by construction of Eq. 6, increasing $\kappa$ does not necessarily mean
increasing the millennial-scale oceanic effect alone, since this would enhance concurrently both the millennial and the orbital-scale components in the ocean. Therefore, investigating the oceanic millennial-scale variability effect on the past GrIS is not as straightforward as expected. Moreover, none of the four parameters of Eq. 6 is perfectly constrained in reality, and a sensitivity study on the influence of their chosen values on the GrIS evolution would be required.

For these reasons, it is first useful to characterise the impact of millennial-scale variability in the ocean on the GrIS evolution
testing a broad range of values of the key-parameters in Eq. 6. Following this approach, we perform a large ensemble (LE) of model simulations using the near-random Latin Hypercube Sampling (LHS) technique (McKay et al., 1979), which allows us to efficiently explore the phase-space of the three parameters minimising the LE computational cost with respect to the full-factorial sampling technique. The LHS technique has already been used to constrain different ice-sheet model parameters and to assess their influence on the model's behavior (Stone et al., 2010; Applegate et al., 2012; Stone et al., 2013; Robinson et al.,
2017). By construction, the $\alpha$ and $\beta$ indices share the same normalisation. Thus the glacial-interglacial and the interstadial-stadial oceanic temperature anomalies have the same (but opposite) amplitudes. This is also supported by estimate of both surface temperature anomalies (Liu et al., 2009; Zhang et al., 2014b; MARGO, 2009; Annan and Hargreaves, 2013). The



problem is therefore reduced to only three degrees of freedom: $\Delta T_{\mathrm{ocn}}^{\mathrm{mil}}$ is set to vary between 0 and 3 K (SST around Greenland deduced from Liu et al. (2009), Zhang et al. (2014b), Vettoretti and Peltier (2015) and Bagniewski et al. (2017)), $B_{\mathrm{ref}}$ between 0 and 10 $m\ a^{-1}$ (chosen as a reasonable climatic mean between Rignot et al. (2010), Straneo et al. (2012), Rignot et al. (2016) and Wilson et al. (2017) for the largest tidewater glaciers around the GrIS, and Rignot et al. (2013) and Liu et al. (2015) for the Antarctic domain) and $\kappa$ between 0 and 10 $m\ a^{-1}\ K^{-1}$ (following Rignot and Jacobs (2002) for the Antarctic domain).

The parameter values are sampled from the specified ranges and, assuming that they are independent from each other, they are randomly combined to generate a total LE of 100 simulations (Fig. 2), named TOT simulations. At the same time, we perform another set of identical simulations, except for the fact that the millennial contribution is set to 0 K for all of them. These are named ORB simulations from now on and are used for direct comparison with the TOT ones, as discussed in Section 3. To initialise the model we use the present-day topography of Greenland from Schaffer et al. (2016). All the simulations of the LE

cover the last two glacial cycles, with the first 120 ka considered as spin up and therefore not analysed.

## 3   Results

### 3.1   Characterisation of oceanic millennial-scale variability effect

A first evaluation of the millennial-scale oceanic variability effect on the whole GrIS can be made looking at the evolution of the simulated ice volume throughout the LGP (Fig. 3a). Apart for low oceanic forcing configurations for which both ORB

(black curves) and TOT (blue curves) present small ice volume deviations from their simulated present-day value, TOT simulations show higher ice volume fluctuations than ORB ones, especially during Marine Isotope Stage 3 (MIS3). The impact of millennial-scale climate variability on the GrIS is better assessed through variations of the GrIS ice volume in TOT away from the orbital-only controlled reference simulation (ORB) (Fig. 3b). This quantity is simply calculated as the difference between the ice volume simulated in TOT and that simulated in ORB for each pair of simulations in the LE. For a higher millennial-

scale variability signal in the ocean, we expect more millennial-scale variability in ice volume. To quantify the contribution of millennial-scale climate variability to the ice-volume variations, we use the root mean square deviation (RMSD) of the residuals between the ice volume simulated in TOT and in ORB (m SLE), defined for the whole time series as:

$$\mathrm{RMSD}_{\mathrm{Vol}} = \sqrt{\frac{1}{\mathrm{N_t} - 1} \sum_{\mathrm{t}=1}^{\mathrm{N_t}} \left(\mathrm{Vol}_{\mathrm{TOT}}(\mathrm{t}) - \mathrm{Vol}_{\mathrm{ORB}}(\mathrm{t})\right)^2} \tag{7}$$

$N_t$ is the total number of time steps in each simulation and $Vol_{TOT}(t) - Vol_{ORB}(t)$ is the ice-volume residual calculated as

the difference between the two simulations at each time step. This quantity tells us how much the millennial volume deviates from the background ORB simulation throughout the LGP. Fig. 3b associates the calculated RMSDs with the millennial-scale contribution to ice-volume variations (m SLE) for each simulation of the LE. The ice-volume contribution due to millennial-scale variability can reach peaks of more than 1.5 m SLE at certain times during the LGP. However, this volume anomaly does not necessarily imply a high millennial-scale ice-volume contribution in terms of the RMSD calculated over the entire time

series.





A second and complementary approach used to identify the effect of the oceanic millennial-scale variability on the GrIS evolution is to calculate the deviation of the ice velocity simulated by TOT from its background state ORB. The methodology is similar to that used for the ice volume, except for the fact that now the standard deviation is first calculated for each grid point $ij$ as $\mathrm{RMSD}_{\mathrm{U,ij}} = \sqrt{\frac{1}{N_t-1} \sum_{t=1}^{N_t} (U_{ij,TOT}(t) - U_{ij,ORB}(t))^2}$ and then averaged for the entire domain:

$$\mathrm{RMSD}_{\mathrm{U}} = \frac{\sqrt{\sum_{ij} \mathrm{RMSD}_{ij}^2}}{N} \tag{8}$$

where N is the total number of grid points of the GrIS domain.

To characterise the resulting effect of the oceanic millennial-scale variability on the GrIS evolution during the LGP, we jointly examine the RMSDs calculated for ice volume and velocities. Ice-volume evolution is a good benchmark for understanding the overall effect of the oceanic variability throughout the time, but U is the direct expression of how the ice dynamics are affected by the forcing. Therefore, we associate the largest millennial-scale variability in the ocean to concomitantly high volume and

velocity RMSDs. The higher the millennial-scale variability in the ocean, the more the ice volume and ice velocity in TOT are expected to deviate from the ORB simulation. Nevertheless, we are interested in understanding which combination of perturbed parameters in the basal melting equation Eq. 6 leads to the highest millennial variability in terms of both quantities. We find that the effect of a high oceanic millennial-scale variability is fairly well constrained in the parameter phase-space, at least per each couple of analysed variables (Fig. 4). Very low oceanic-driven variability is generally associated with low values of $\kappa$

and $\Delta T_{\mathrm{ocn}}$ (weak oceanic forcing). These configurations lead to basal melting evolutions that do not deviate much from the reference-state basal melting throughout the LGP (blue curves of Fig. SM1 in Supplementary Material). Low millennial-scale variability is also found for combinations of low $B_{\mathrm{ref}}$ with high $\kappa$ (or high $\Delta T_{\mathrm{ocn}}^{\mathrm{mil}}$), and for high $\kappa$ with high $\Delta T_{\mathrm{ocn}}^{\mathrm{mil}}$. The reason is that these are associated with a basal-melt evolution that rapidly saturates at the cut-off value $0~m~a^{-1}$ (no melting, since freezing is not allowed) after the Eemian and maintains that state throughout the LGP (light blue curves in Fig. SM1 in

Supplementary Material). It is between these two configurations, with medium-high $B_{\mathrm{ref}}$ and high $\kappa$, that the resulting basal melting signal allows for a high millennial-scale variability in the ocean, i.e. a signal sprinkled with sufficiently high melting peaks at millennial time-scales that sometimes saturates at $0~m~a^{-1}$. Similar logic can be followed considering $\Delta T_{\mathrm{ocn}}^{\mathrm{mil}}$ instead of $\kappa$. Therefore, high $\kappa$ must be associated with low $\Delta T_{\mathrm{ocn}}^{\mathrm{mil}}$ and vice versa to produce high millennial-scale variability in the basal melt, as reflected in Fig. 4 as well.

## 3.2    Oceanic millennial-scale variability impact on transient dynamics

The aim of this section is to investigate the effect of oceanic millennial-scale variability on the transient GrIS dynamics and ice-sheet evolution during the LGP. To this end, we choose the simulation from the LE which corresponds to the maximum millennial-scale variability response in terms of both ice volume and velocity and we compare it to its corresponding background simulation. As expected from the characterisation of the millennial variability discussed above, the simulation chosen

has medium-high $B_{\mathrm{ref}}$ ($7.6~m~a^{-1}$), high $\kappa$ ($8.3~m~a^{-1}~K^{-1}$) and consequently medium $\Delta T_{\mathrm{ocn}}^{\mathrm{mil}}$ ($1.2~K$). From now on, this simulation and its corresponding orbital-only reference simulation are referred to as $\mathrm{TOT}_{\mathrm{max}}$ and $\mathrm{ORB}_{\mathrm{max}}$, respectively.





We compare ice thickness, velocity and migration of the grounding-line position simulated by $TOT_{max}$ with its corresponding fields simulated at the same time in the $ORB_{max}$ simulation throughout the whole LGP (Movie in Supplementary Material). Note that velocity is compared between two different GrIS extensions, thus a velocity difference in regions that show a mismatch in ice cover corresponds to the velocity of the simulation in which the grounding line has not retreated yet. Since the millennial-scale $\beta$ index is built from the NGRIP-temperature reconstruction through filtering to retain millennial timescales, each of the submarine melting peaks is associated with a D-O event culminating in a Greenland Interstadial (GI) period. We show the comparison specifically for the time period corresponding to GI 16 (ca. 56.8 ka BP) associated with one of the pronounced peaks in the basal melting fluctuation during the MIS3 (Fig. 5). In the $TOT_{max}$ simulation a large ice thickness decrease is found in the NEGIS (NE Greenland ice stream) and the Kangerdlugssuaq fjord (SE Greenland) regions (Fig. 6). These ice reductions are associated with a large grounding-line retreat during GI 16. The region near Baffin Bay (NW Greenland) is also affected by inland grounding-line migration, however the retreat is limited if compared to the other regions. The ice stream related to Jakobshavn Isbrae outlet glacier in SW Greenland shows increased velocities, leading to increased ice discharge. During the slow cooling after the peak of the GI (at ca. 56 ka BP), the grounding line advances back, but the ice recovery is limited. The $ORB_{max}$ simulation shows very little grounding-line and ice-thickness responses for the whole analysed time period (Movie in Supplementary Material). For example, increased velocities associated with enhanced ice discharge lead to a large grounding-line retreat on the western margin of the GrIS during GI 12 (ca. 45.8 ka BP) when millennial-scale variability in ocean is included, and small ice-thickness reductions and grounding-line retreats are also simulated around NEGIS. The regions around Kangerdlugssuaq fjord and Sermilik fjord (SE Greenland) experience a large decrease in ice thickness linked to higher velocity and ice discharge throughout the peak. Appreciable grounding-line variations induced by millennial-scale variability in the ocean are also observed during the GI 8 (ca. 37.4 ka BP). Considerable grounding-line retreat and associated ice discharge are found close to Baffin Bay. The Kangerdlugssuaq region shows high velocities and an ice-thickness decrease, but this is related to only a minor grounding-line migration. The simulated NEGIS area does not show appreciable response to the submarine melting peak, since the grounding-line position reached by both simulations before the peak is maintained throughout the GI. As for GI 16, the $ORB_{max}$ simulation does not show a substantial ice and velocity change, except for the strong velocity increase and ice-thickness reduction in Jakobshavn outlet glacier. This is related to the large ice thickness simulated in the 10 ka previous to GI 8, which increases the amount of glacial water produced at the base of the ice stream, enhancing basal sliding and ice discharge. On the other hand, the presence of millennial-scale oceanic variability allows for frequent ice discharge from the Jakobshavn glacier, limiting the basal-water amount and promoting ice growth.

Generally, $ORB_{max}$ is much more static than $TOT_{max}$, showing smaller grounding-line and ice-discharge changes. Including millennial variability in the ocean leads to variations in ice-thickness of hundreds of meters with respect to the corresponding orbital-only case in locations close to the marine margin. This difference is not only exhibited at the ice-ocean front, but it also penetrates through the interior of the ice sheet by several tens of kilometers. This effect is related to the propagation of velocities, which, especially around the west GrIS, Sermilik, Kangerdlugssuaq and NEGIS regions, can penetrate far inland promoting ice discharge from the interior.



The transient effect of the oceanic millennial-scale variability throughout the ice sheet can be seen by analysing the LGP evolution of ice thickness and velocity for a single grid point of the domain, at three strategic locations: close to the glacial marine margin (A), close to the PD marine margin (B, glacial ice-sheet interior) and far inland in the ice sheet (C). We show the results of this analysis for the Baffin Bay region (Fig. 7), but other two areas around the Greenland domain have been investigated (Fig. SM2 for Jakobshavn Isbrae and Fig. SM3 for the NEGIS, in Supplementary Material). In the region close to

Baffin Bay intermittent periods of high submarine melt lead to high velocity fluctuations ($600-800\ m\ a^{-1}$) limiting ice growth at the glacial ice margin. For more than 40 ka during the LGP, the ice thickness in the $\mathrm{TOT_{max}}$ case (blue curve) is below that produced in the $\mathrm{ORB_{max}}$ simulation (black curve) by 300-400 m, reaching melting peaks in which ice completely disappears for more than 4 ka. A similar ice-thickness evolution is found in the proximity of the PD marine margin, about 160 km far from the glacial ice-ocean border (point B). The effect of the millennial variability in the ocean propagates inland, preserving

an ice thickness increase of 500 m there and by 200 m even 300 km away from the ocean border (point C), despite minor or no observable difference in velocities. These estimates further corroborate the results found for the Baffin Bay region in the 2D plots (Movie in Supplementary Material). The region including the Jakobshavn Isbrae (Fig. SM2) seems to be less affected by the millennial fluctuation in basal melting than the region of Baffin Bay. Still, some ice-thickness variations are found close to the marine margin, especially around 50-40 ka BP, when the ice shelf repeatedly disappears. Little evidence of this ice

fluctuation is found in the ice interior, however, although decreased velocities with respect to the $\mathrm{ORB_{max}}$ simulation around 40 ka BP lead to ice growth culminating in ice 700 m higher than that of the $\mathrm{ORB_{max}}$ simulation (point B). The ocean-induced variability rapidly decays further inland (point C). However, it must be noted that this latter location is more than 600 km away from the marine margin, thus this attenuation is expected. Millennial-scale fluctuations in submarine melting strongly impact the glacial evolution in the NEGIS region (Fig. SM3) and very high velocity fluctuations ($1000-3000\ m\ a^{-1}$) between 50-30

20     ka BP constrain ice growth offshore (point A). The absence of ice at location A precludes the buttressing effect that limits ice discharge from the ice-sheet interior, as seen in the $\mathrm{ORB_{max}}$ simulation. On the contrary, the presence of millennial-scale fluctuations in the ocean helps to maintain ice advection from the interior toward the ice-ocean margin, favouring ice discharge from the interior and limiting the ice increase by 1000 m (point B) and 500 m far inland (point C).

The effect of millennial-scale variability in the ocean throughout the LGP is summarised in Fig. 8 at large spatial scale. Ice

thickness and velocity RMSDs both calculated following Eq. 8 for the whole time period show that regions that exhibit a strong response to the ocean in terms of ice thickness variations are also subjected to strong changes in velocity. This is highlighted in regions such as Baffin Bay, Jakobshavn Isbrae, Kangerdlugssuaq and Sermilik fjords and NEGIS, confirming the results discussed in this Section for specific times and locations.

## 4    Discussion

### 4.1    Comparison of model results to proxy data

Our simulations show that small temperature changes can result in significant ice-volume fluctuations from the background glacial GrIS configuration. Also, the GrIS may have contributed to IRD discharge during the LGP, as suggested by proxy





records (Andrews et al., 2012) and it is therefore interesting to outline its possible imprint associated with millennial-scale variability in the ocean.

We compare our model results to proxy data taken from sediment cores drilled in locations that could have been partly affected by recurrent ice discharge from the GrIS throughout the LGP. The locations are situated in the Labrador Sea (MD95-2024, (Weber et al., 2001)), in the northwestern margin of Iceland (PS2644-5, (Voelker and Haflidason, 2015)) and in the

North Atlantic along the eastern side of the Reykjanes Ridge (JPC-13, (Hodell et al., 2010)), close to the so-called IRD belt (Ruddiman, 1977). For each analysis we show the ice flux averaged over the closest coastal zone in Greenland, and simulated by $\text{TOT}_{\text{max}}$ (solid line) and by $\text{ORB}_{\text{max}}$ (dashed line), compared to a specific proxy data for the LGP (Fig. 9 - 11). Gamma ray density ($g\ m^{-3}$) reconstructed from sediment core MD95-2024 (Labrador Sea, Fig. 9) is compared to the simulated ice flux averaged over the Baffin Bay region. All main Heinrich Events (HE) and some of the GI of the LGP can be recognized in

the gamma ray density signal (Weber et al., 2001), which follows the IRD data extracted from the same sediment core. A weak correlation between model and data is found, especially for GI 16, GI 12 and GI 1. The most important source of IRD found in the Labrador Sea is thought to be Hudson Bay, which is likely responsible for the majority of iceberg discharge during HEs. However, some of the D-O events found in that region have been attributed to ice discharge from Baffin Bay (Andrews et al., 2012) and our results seem to partly follow this hypothesis. The total lithic fragment (IRD and thepra grains) extracted from

sediment core PS2644-5 (NW Iceland, Fig. 10) is compared to the simulated ice flux averaged over the ice-covered Denmark Strait area close to Scoresby Sund. Peaks GI-21, GI-18, GI-12, GI-8 and GI-1 are visible in the modeled ice flux, showing a good correspondence between ice flux from the East GrIS and IRD deposition. Thus millennial-scale variability in the ocean could be partly responsible for enhanced iceberg production from East-Southeast GrIS bringing IRD to the PS2644-5 site by the East Greenland Current (Voelker and Haflidason, 2015). IRD data (% of lithic fraction) from sediment core JPC-13 in

southern Gandar Drift in the North Atlantic (Fig. 11) are compared to the simulated ice flux averaged over the glacial marine-based GrIS area between Kangerdlugssuaq and Sermilik Fjords. Some agreement between modeled ice flux and lithic fraction is found for HE-1, HE-3, HE-4, HE-5, HE-6. However, the model seems to lag each HE by 2-3 ka, suggesting that these ice flux peaks are likely a response to D-O events as simulated in the submarine melting rather than to the observed HEs. The lack of temporal correspondence between model results and proxy data can be explained considering that the IRD found in JPC-13

may have come from other NH ice sheets and transported by icebergs via the stronger Iceland-Scotland current during MIS3 (Hodell et al., 2010). On the contrary, icebergs discharged from SE Greenland may likely be transported far from the IRD belt region towards the Labrador Sea. Another possible explanation for this low model-data correspondence is related to the usage of SSTs to force the ocean, which results in ice discharge peaking in phase with the NGRIP-derived atmospheric temperatures. On the contrary, ice discharge during HEs is thought to be related to subsurface waters, which are supposed to be in antiphase

with respect to the atmosphere, thus leading to warmer ocean temperatures during stadials (Marcott et al., 2011; Alvarez-Solas et al., 2013). It is therefore possible that considering oceanic temperatures at other depths in the water column could improve the synchronization between modelled and reconstructed IRD discharge during HEs.

Despite individual disagreements, all of these comparisons show that a more responsive GrIS, as obtained when the model is forced with millennial-scale oceanic variability ($\text{TOT}_{\text{max}}$), is in better agreement with proxies compared to the orbital-only



simulation (ORB$_{\mathrm{max}}$). Rapid (millennial-scale) basal melting fluctuations allow for ice-flux increases matching the timing of some of the peaks of the proxy data, meaning that the GrIS could have contributed to the ice discharge at those times. With this analysis, we do not aim to precisely reconstruct the timing and spatial distribution of the iceberg discharge during D-O events. Instead, we aim to explore the implications of considering the oceanic millennial-scale variability on the GrIS ice fluctuation during the LGP. It is surprising how much the sole presence of millennial-scale variability in the ocean can influence the GrIS

evolution and ice discharge during the LGP interstadials, notwithstanding that millennial-scale variability is neglected in the atmosphere. However, tracing the origin of IRD deposition is a complex and open problem that needs a deeper understanding of changes in the ocean column during D-O events, as well as a better knowledge of the oceanic and glaciological processes that ultimately determine the deposition of IRD on the ocean floor due to iceberg discharge.

### 4.2 Model limitations and caveats in basal melting parameterisation

Some of the GRISLI-UCM model limitations already discussed by Tabone et al. (2018), such as the coarse model resolution (20 km by 20 km), which impedes the correct solution of small and steep fjords, the relatively simple GIA scheme, which takes into account only local changes in the ice load, and the PDD ablation scheme, which does not consider changes in past insolation, are also present in this work.

In the basal melting rate parameterisation we have introduced a cutoff value to prevent freezing at the ice-shelf base and thus

limit ice accretion during the glacial period. This assumption is reasonable for the low spatial resolution of the model, however, it ignores an existing process, since freezing is observed at the base of many marine glaciers, even for the present day (Rignot et al., 2013).

The parameters $B_{\mathrm{ref}}$, $\Delta T_{\mathrm{ocn}}^{\mathrm{mil}}$ and $\Delta T_{\mathrm{ocn}}^{\mathrm{orb}}$ in the basal melting equation are assumed to be spatially uniform. We are aware that spatially-variable fields should be taken into account for an exhaustive investigation of the problem. Both stadial-interstadial

and glacial-interglacial temperature anomalies could be taken from existing transient model outputs, for instance. However, a complete map of the observed PD basal melting rates for the whole Greenland domain does not exist yet, in contrast to Antarctica (Rignot et al., 2013), thus limiting the effectiveness of including additional complexity at this time.

The indices $\alpha$ and $\beta$ used to create the past orbital and millennial forcings, in particular for the submarine melting, are built based on the NGRIP-derived temperature signal. As far as we know, no high-resolution ocean temperature reconstructions

covering the entire LGP exist for the North Atlantic. However, our assumption that the first layers of the ocean vary in phase with the atmosphere is reasonable and using the same climatic indices for the ocean is thus a fair approximation to assess the problem. By allowing the glacial-interglacial and stadial-interstadial anomaly temperatures to range between 0 and 3 K, we are considering basal melting rates at the grounding line and below the ice shelves as driven by SST variations (MARGO, 2009; Liu et al., 2009; Annan and Hargreaves, 2013; Bagniewski et al., 2017; Jensen et al., 2018). However, grounding lines of many

outlet glaciers in Greenland are estimated to be located deeper in the water column (Wilson et al., 2017). Therefore subsurface temperature anomalies should be considered in these cases, which may lead to slightly different results, since they are probably in antiphase with the atmosphere (Zhang et al., 2014b; Vettoretti and Peltier, 2015). However, as a sensitivity study we do not aim to perfectly reproduce reality but rather analyse the possible effects of an ocean driven by millennial-variability on the

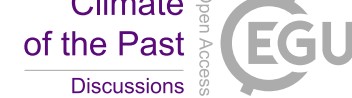



ice-sheet evolution in Greenland. This assumption, combined with the use of a linear basal melting parameterisation based only on temperature variations without refreezing, lead us to ignore more complex mechanisms that occur at the ice-ocean bed interface, such as meltwater buoyancy and water convection in the ice-shelf cavity (Jenkins, 2011). However, in the framework of a sensitivity study these limitations should not influence the main objective of this work.

The basal melting evolution reproduced in the $\mathrm{TOT}_{\mathrm{max}}$ simulation (solid line in Fig. 5) is strongly related to a specific

combination of the perturbed parameters in the basal melting equation (Eq. 6). This combination is reasonably chosen from the range of possible values as it produces the maximum oceanic millennial variability in the LE, but the same analysis could be performed for other basal melting configurations. A different magnitude of the melting peaks during MIS3 (Fig. SM1) or a different distribution of the melting peaks (resulting from the use of a different temperature reconstruction to create the $\alpha$ and $\beta$ indices) may vary the timing of grounding-line advance and retreat or ice growth and loss. This is also true for the

orbital-driven basal melting forcing applied to the $\mathrm{ORB}_{\mathrm{max}}$ simulation (dashed line in Fig. 5), which shows submarine melting rates higher than $0\ m\ a^{-1}$ for only 10 ka during MIS3 (between 55 and 45 ka BP). A different melt configuration could vary the timing and spatial distribution of grounding-line migration and ice cover.

Finally, millennial-scale variability in the atmosphere is not considered here. This simplified experimental design is chosen following previous results demonstrating the important role of the ocean in the GrIS evolution over the last glacial periods at

orbital timescales (Tabone et al., 2018). The experiments here allow us to directly investigate how millennial-scale variability in the ocean can impact the evolution of the GrIS. However, we have omitted an important component of the climate system and to comprehensively understand the effect of millennial-scale fluctuations on Greenland, further experiments should be done by including this variability in air temperature and precipitation.

## 5   Conclusions

We have assessed the effect of the millennial-scale oceanic variability on the evolution of the GrIS during the Last Glacial Period. To do so, we used an ice-sheet-shelf model, in which the millennial-scale variability in the ocean was imposed as a fluctuation in the basal melting rate at the grounding line and below the ice shelves. We first characterised the millennial variability through a sensitivity test for a broad range of values of the perturbed parameters in the submarine melting equation. We showed that the millennial-scale contribution to ice-volume variations during the LGP could have reached peaks of more

than 1.5 m SLE. The southeastern area around Kangerdlugssuaq fjord, Baffin Bay and the NEGIS regions were found to be very sensitive to millennial-scale variability in the ocean. Ice thicknesses simulated at the marine margin differed by 500-1000 m from that simulated by orbital-only driven oceanic variations. Moreover, imprints of these differences are still found for several tens (hundreds - in certain regions) of kilometers far from the ice-ocean interface due to the velocity-driven upstream propagation of the ice-flow perturbation. Although the aim of this work was far from assessing the true timing and spatial

distribution of any GrIS ice discharge that occurred during the D-O events, we showed that considering the millennial-scale variability in the ocean is necessary to reproduce some of the IRD peaks observed in North Atlantic proxy data. Our work thus suggests that millennial-scale induced changes in ocean circulation and temperature may be important drivers of the GrIS





evolution during the LGP, advancing the hypothesis of a potential role of the GrIS in oceanic reorganisations at millennial timescales.

*Code availability.* The GRISLI-UCM model code is available from the authors upon request.

*Author contributions.* IT performed the simulations, analysed the results and wrote the paper. All the authors of this work contributed to conceive the experiment and writing the manuscript.

5   *Competing interests.* The authors declare that they have no conflict of interest.

*Acknowledgements.* The work was supported by the Spanish Ministry of Science and Innovation in the framework of the project MOCCA (Modelling Abrupt Climate Change, grant no. CGL2014-59384-R). Ilaria Tabone is funded by the Spanish National Programme for the Promotion of Talent and Its Employability through grant no. BES-2015-074097. Alexander Robinson is funded by the Ramn y Cajal Programme of the Spanish Ministry for Science, Innovation and Universities. The model simulations were carried out in the HPC of Climate Change of
10  the International Campus of Excellence of Moncloa (EOLO), supported by MECD and MICINN. Finally, we are thankful to Catherine Ritz for providing the original model GRISLI.



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





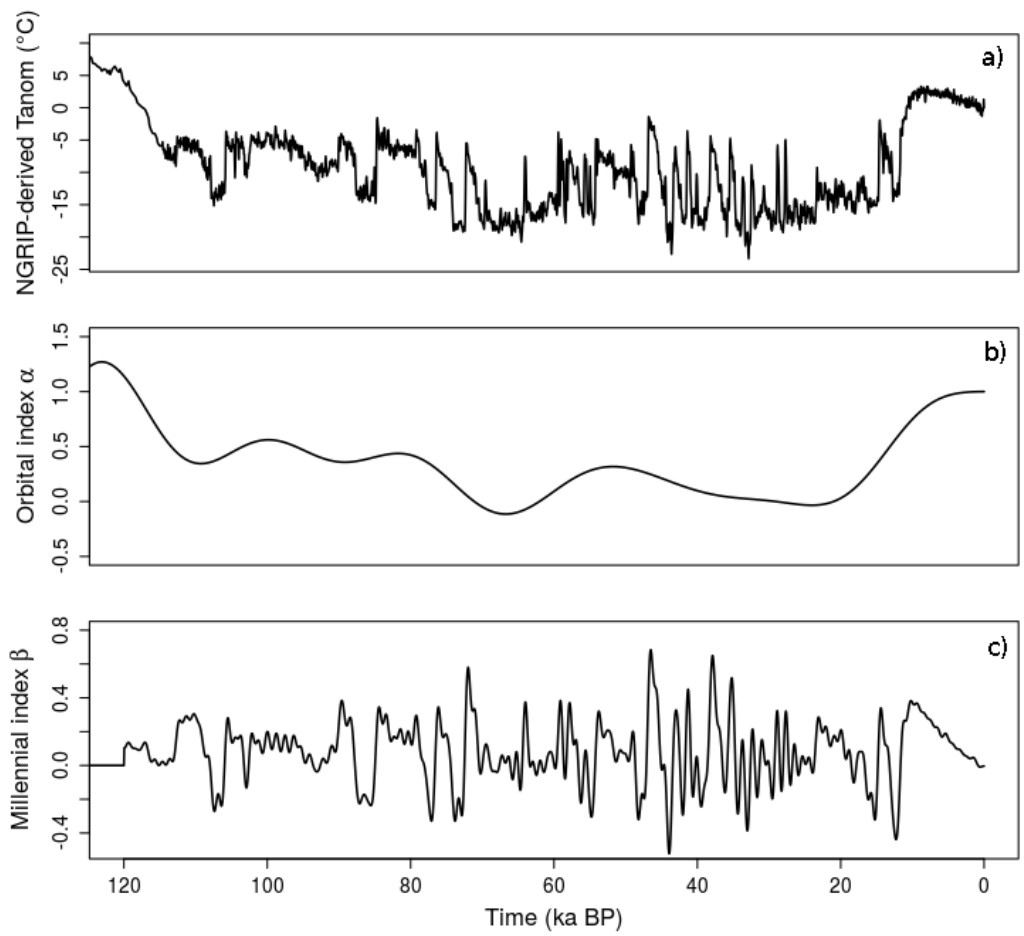

**Figure 1.** NGRIP-Temperature anomaly reconstruction (a) used to construct the orbital-index $\alpha$ (b) and the millennial-index $\beta$ (c) for the last glacial period. $\alpha(t)$ is built from the NGRIP-derived temperature anomalies (Tabone et al., 2018), filtered to remove the spectral components below the orbital period ($1/f < 18\,ka$) and normalized between 0 and 1. $\beta$ is derived by subtracting the orbital $\alpha$ index from the NGRIP-derived temperature signal, normalized between 0 ad 1, and then filtered below 1 ka to remove sub-millennial periods.



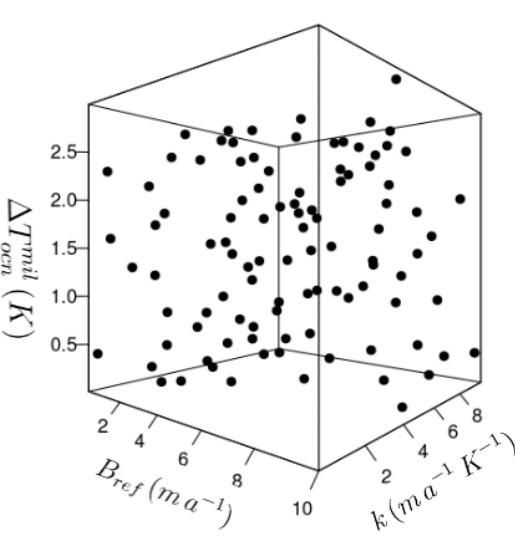

**Figure 2.** Distribution of of LE simulations produced by the Latin Hypercube Sampling (LHS) technique. The phase-space of parameters is built from $B_{\mathrm{ref}}$ ranging from 0 to 10 m a$^{-1}$, $\kappa$ from 0 to 10 m a$^{-1}$ K$^{-1}$ and $\Delta T_{\mathrm{ocn}}^{\mathrm{mil}}$ from 0 to 3 K.





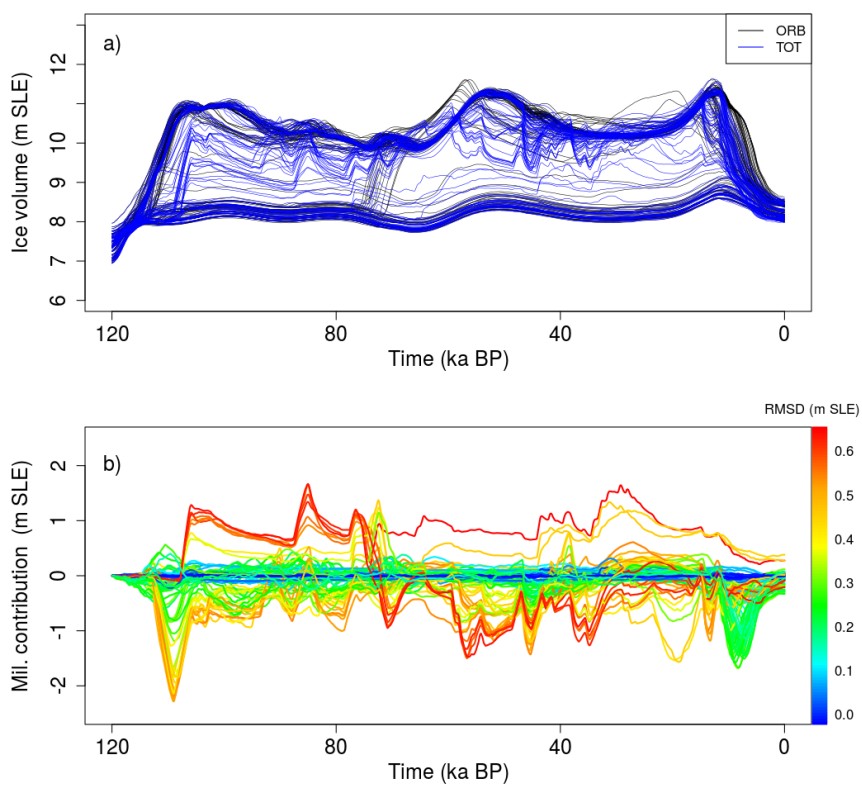

**Figure 3.** a) Ice volume evolutions simulated by ORB (black curves) and TOT (blue curves) throughout the LGP (m SLE) b) Oceanic millennial-scale variability contribution to the GrIS ice volume variation (m SLE; positive values indicate ice loss to the ocean). The color scale indicates the RMSD calculated in terms of ice volume residuals between TOT and ORB simulations (m SLE).



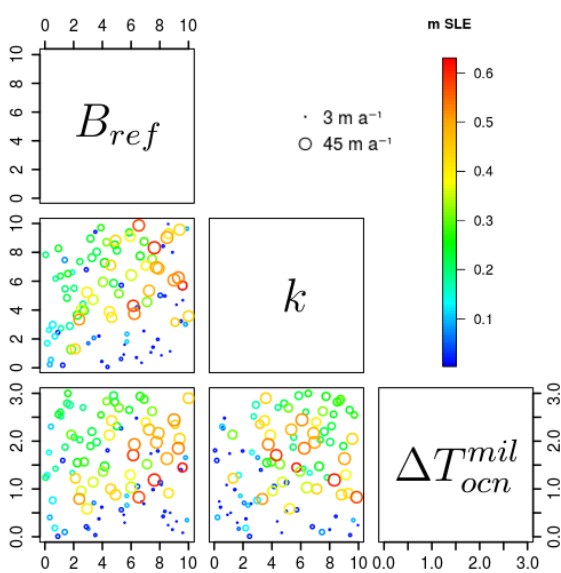

**Figure 4.** Evaluation of the effect of the oceanic millennial-scale variability for each pair of parameters perturbed in the basal melting equation (Eq. 6). The range volume (m SLE, colour scale) and ice velocity ($m\ a^{-1}$, circle size) RMSDs is shown. The simulation with the highest millennial-scale variability is represented by the largest red circle; the simulation with the lowest variability is represented by the smallest blue circle.





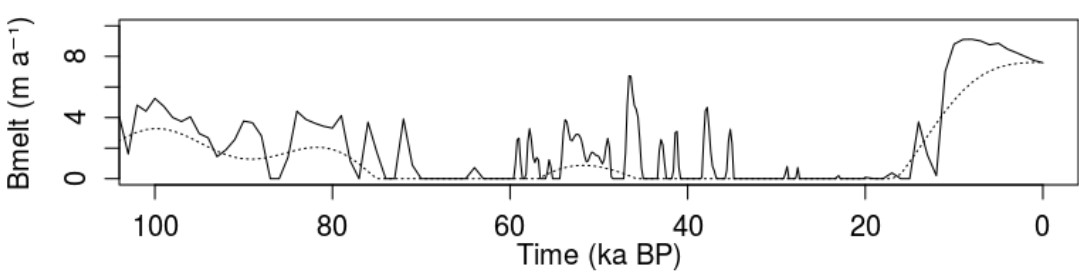

**Figure 5.** Evolution of the submarine melting used to force the $\mathrm{TOT_{max}}$ (solid black line) and $\mathrm{ORB_{max}}$ (dashed black line) simulations for the last 100 ka.



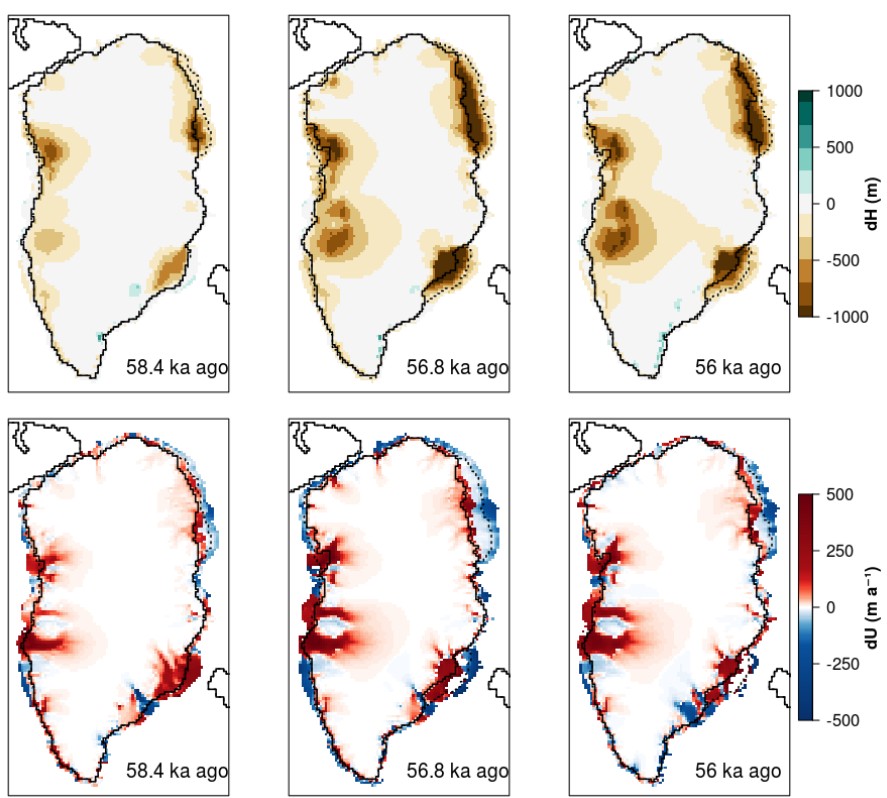

**Figure 6.** $\mathrm{TOT_{max}}$ - $\mathrm{ORB_{max}}$ ice thickness (m, upper panels) and the corresponding $\mathrm{TOT_{max}}$ - $\mathrm{ORB_{max}}$ velocities ($m\,a^{-1}$, lower panels) for three different times during submarine melting rate peak corresponding to GI 16. The position of the grounding line in the $\mathrm{TOT_{max}}$ and $\mathrm{ORB_{max}}$ simulations are indicated by solid and dashed black contour, respectively.



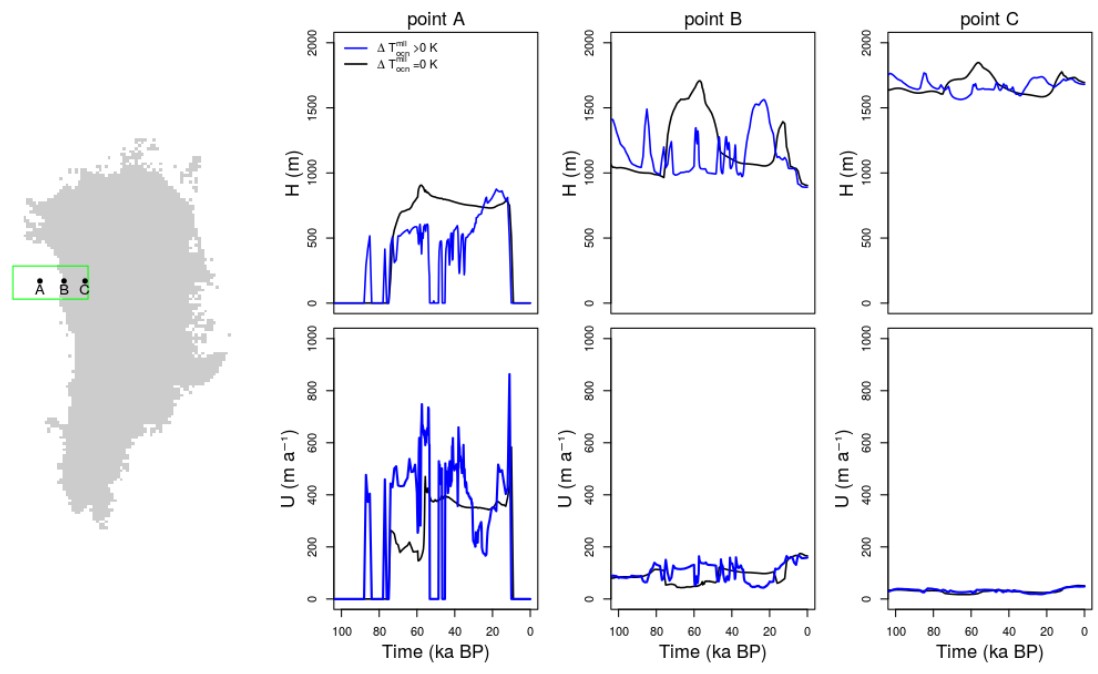

**Figure 7.** Transient dynamics specified for three locations along Baffin Bay. Ice thickness (upper panels) and velocity (lower panels) LGP evolution is shown for the glacial marine margin (point A), present-day marine margin (point B) and far in the interior of the ice sheet (point C). Black curves indicate the dynamics of the $\mathrm{ORB_{max}}$ simulation, blue curves represent the dynamics of the $\mathrm{TOT_{max}}$ simulation.

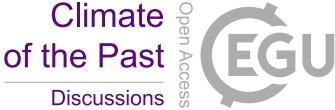



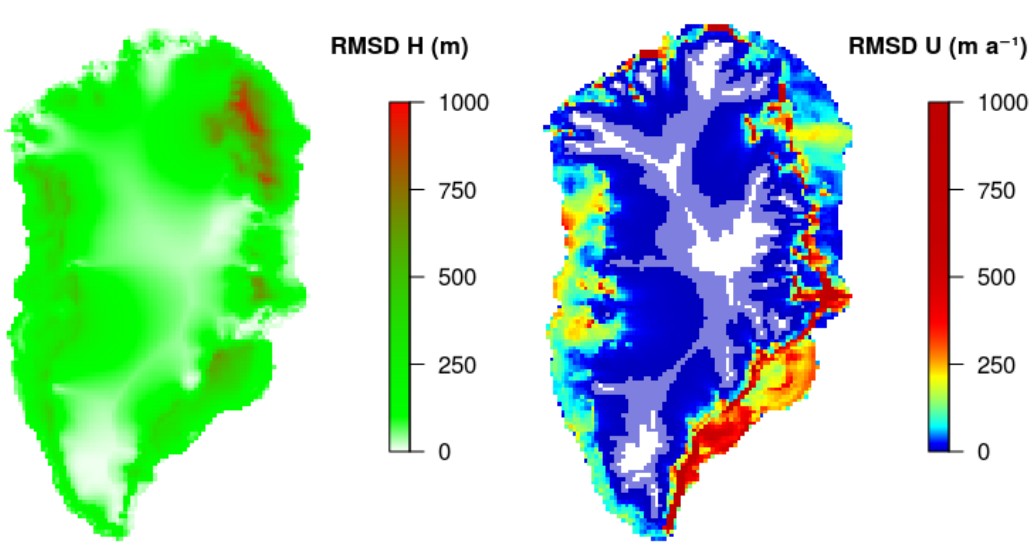

**Figure 8.** RMSD for ice thickness H (m) and velocity U $(\mathrm{m\,a^{-1}})$ calculated between $\mathrm{TOT_{max}}$ and $\mathrm{ORB_{max}}$, following Eq. 8.





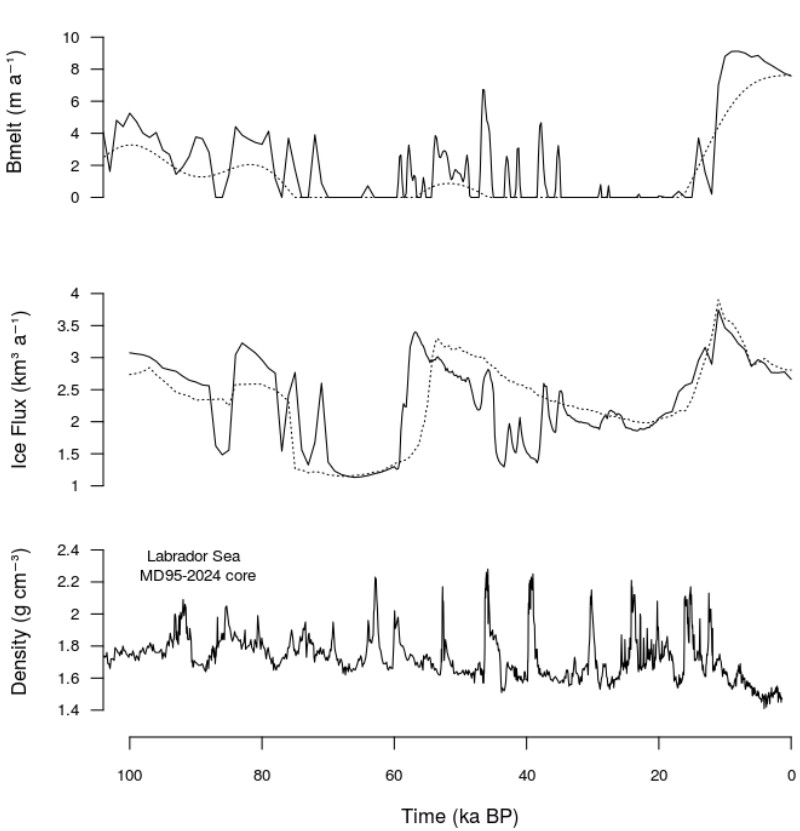

**Figure 9.** Simulated submarine melt (upper panel) and simulated ice flux averaged over the Baffin Bay region (middle panel) are compared to proxy-derived gamma ray density from the sediment core MD95-2024 in the Labrador Sea (lower panel).

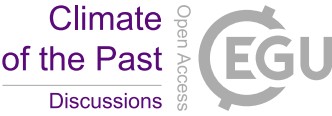



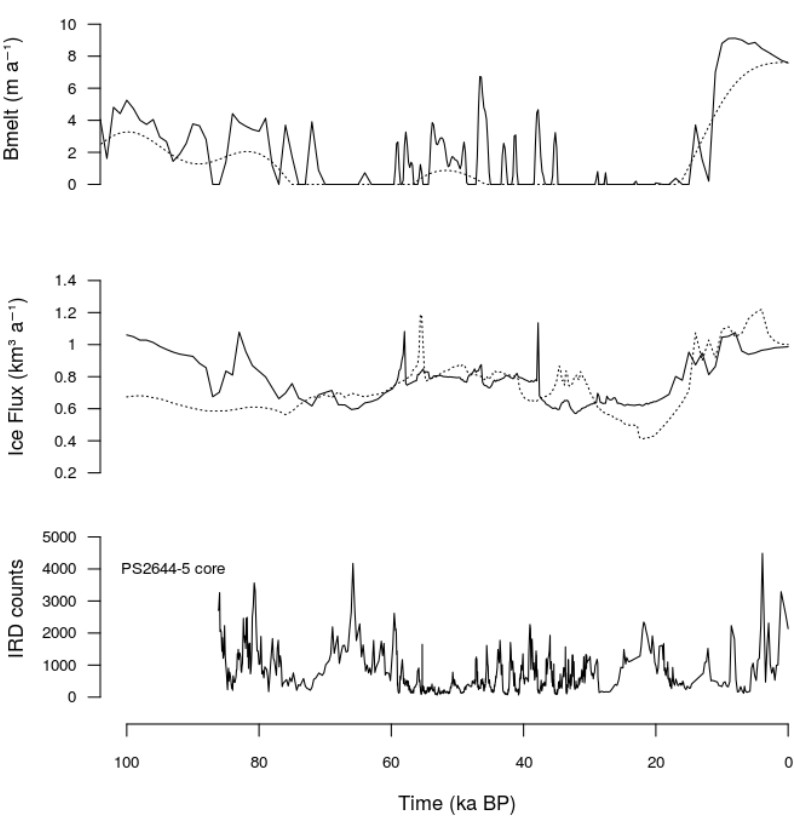

**Figure 10.** Simulated submarine melt (upper panel) and simulated ice flux averaged over the Northern part of Denmark Strait (middle panel) are compared to the total lithic fragments extracted from the sediment core PS2644-5 close to the North West Iceland (lower panel).





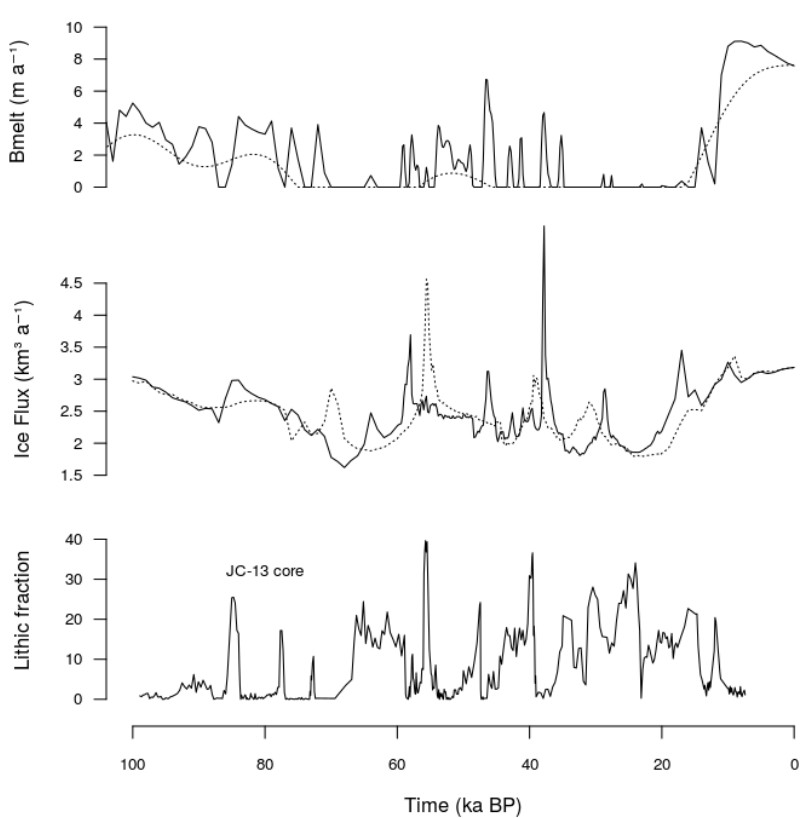

**Figure 11.** Simulated submarine melt (upper panel) and simulated ice flux averaged over the South East GrIS region (middle panel) are compared to the percentage of total lithic fraction induced by the JPC-13 sediment core in southern Gandar Drift in the North Atlantic (lower panel).