# Peer review of "Impact of millennial-scale oceanic variability on the Greenland ice sheet evolution throughout the Last Glacial Period"

_Climate of the Past, 2018_

## Referee Comment (RC1) · A. Svensson (Referee) · 12 Oct 2018

The manuscript (MS) aims at estimating the Greenland contribution to sea level variability during the last glacial period in relation to D-O and Heinrich events. The authors apply an approach similar to that used in an already published paper concerned with glacial-interglacial cycles.

Overall, I think the exercise is very useful to give a first order estimate for the Greenland contribution to sea level variability during glacial times, and I think the approach of implementing ice-shelf ocean interaction is new and very relevant. I have one major point of concern, however, that I think needs consideration in order for the MS to make

a good case.

It is very well established in the literature that during the last glacial period, the Heinrich events and the major ice rafting events in the North Atlantic are associated with the Greenland cold phases, the stadials. In contrast, during the mild Greenland periods, the interstadials, ice rafting and transport of continental material to the North Atlantic is much more limited. This is clearly illustrated in Figure 11 in the cited Hodall et al., QSR, 2010 paper, but also in many other studies where record resolution allows for a detailed comparison of marine records to the Greenland temperature record.

If I understand the model of the manuscript correctly, the ocean-terminating melt of GRiS is forced by the Greenland derived surface temperature, and the ocean temperature variability is assumed to be in phase with the (Greenland) atmosphere. Therefore, most of the marine-induced/basal melt from GRiS occurs during the interstadials. I see this as a highly unrealistic approach. Whereas the Greenland surface temperature is quite likely to give a good estimate for the Greenland surface melt during D-O events, I think this approach leads to a very unrealistic scenario for the basal melt that is mainly caused by interaction between the ice sheet and the subsurface ocean.

As seen in the MS figure 11, the modelled basal melt is completely out of phase with the melt events observed in the marine record. All the major modelled melt events occur during the Greenland interstadials, whereas all of the observed melt events occur during stadials. It is argued in the MS that the reason for this discrepancy could be that the source area for the marine IRD in this specific core could be different from Greenland. However, the timing of other IRD sources are also consistently occurring in the stadials. I think that the reason for this significant model-data disagreement is that the model approach of forcing the basal melt by Greenland surface temperatures is fundamentally wrong.

One could argue that is may not matter so much exactly when the ice sheet is losing or gaining mass as long as the inferred change in sea level variability is of the right

order of magnitude. I this case, however, I think it is quite clear that the GRiS mass changes observed in the model are caused by an entirely wrong mechanism, and therefore are likely to be misleading and possibly of the wrong magnitude, also when it comes to sea level variability. Therefore, I think it is very important that a more realistic approach is applied for the basal melt/marine-terminating ice melt. It can possibly still be a simple approach, but it needs to include some estimated extent of sea ice in the North Atlantic, and some marine-based estimate of sub-surface ocean temperatures. The inclusion of sea ice is essential, because the sea ice partly seals off the ocean from direct heat exchange with the atmosphere, and thereby hampers the assumption of an in-phase temperature variability of ocean and atmospheric temperatures on the time scales relevant for D-O events.

Possibly, the authors can seek inspiration for a more realistic basal melt timing in those papers:

Dokken et al., Paleoceanoraphy, 2013: Dansgaard-Oeschger cycles: Interactions between ocean and sea ice intrinsic to the Nordic seas

Bassis et al., Nature, 2017: Heinrich events triggered by ocean forcing and modulated by isostatic adjustment.

---

## Referee Comment (RC2) · Anonymous Referee #2 · 16 Nov 2018

Tabone et al. use a three dimensional hybrid ice-sheet-shelf model forced by oceanic fluctuations derived from paleo records. The work studies the role of millennial scale climate variability in ocean-ice sheet interaction and is a topic that is of great interest to the glacial climate variability. These type of studies provide a basis to test our earth system models under past climate conditions in order to validate projections of anthropogenic climate change or to develop a better process understanding of climate components that are critical for assessing future anthropogenic climate change.

The introduction provides a nice review of the literature on the subject and reviews the key research questions on ocean-ice sheet interaction. The major problem problem I

see with the ocean-ice sheet coupling is that the oceanic forcing is not realistic and the study would be better suited to investigate the sensitivity of the model to marine shelf instability as a result of sub-surface temperature variations. Page 11, Line 22 states that the model seems to lag Heinrich Events by 2-3 ka. The manuscript goes to great length to compare the model with sediment cores when the forcing (Figure 1c) is representative of surface temperature changes. The authors state that (Page 12, L2) that they do not aim to precisely reconstruct the timing and spatial distribution of ice discharge during D-O events. I don't think it is satisfactory to show that oceanic millennial scale variability is influencing the GrIS evolution alone unless something can be said about a process based explanation of what is happening in the real climate system based on the modelling results.

Other comments:

Section 2.1 Page 4, Lines 15-31: The description of coarse grid points is not clear here. In line 30, the ice sheet model is given (20x20km) but the ocean resolution forcing is not described clearly.

Section 2.1 and 2.2: P4, L31-32; P5 L2; P13 L13 The statement "we consider the atmosphere as modulated by orbital changes". This statement should be rewritten to downplay the impression that there is an atmosphere in the model. Statements such as L31-32 that describe millennial variability in the atmosphere are also misleading. One mainly thinks of atmospheric dynamics in terms of atmospheric climate/weather variability which happens on short timescales. Trace gases, insolation and other slowly evolving atmospheric properties are the result of the internal and external forcing of the earth system.

Section 2.3 P5 L26: "... changes in ocean temperature into..."

P6 L4-7: The construction of beta is a nice measure of millennial scale variability. But if you really want to influence the basal melt rate during the glacial as proposed by some of the studies in the introduction in a realistic fashion the millennial index in Figure 1c

should be inverted to reflect changes in subsurface temperature during Heinrich and D-O stadials. This is a major problem with the study. There is some of this discussion in Section 4.2 on Model limitations and caveats but this detracts from realism of the science and what the study can actually say about what processes are important for millennial scale variability.

P6 L7-8: What is meant by the statement that both deltaTorb,ocn and deltaTmil,ocn are both assumed to be in phase with the atmosphere when there is no deltaTmil,atm in equation (1) and (2)

P6 L12: Adding a short description of how changes in RSL on the orbital timescale are prescribed might be more helpful instead of just the reference.

Section 2.4 P6 L15-20: The authors state that the basal melt is dependant on 4 parameters. Another problem I see is in the variation of the parameter changes during the LHS sampling. The reference basal melt is given as Bref =kappa(Tclim,ocn - T_f) where T_f is fixed. Equation (6) has B proportional to kappa*deltaTorb,ocn. So variations in kappa and Bref are not independent. Changes in kappa will make inverse changes in Tclim,ocn (the mean climatology of the ocean) if Bref is varied in an inconsistent manner. So am I missing some understanding of the variational procedure or is the LHS sampling (which considers previous choices) taking care of this discrepancy? Again in section 2.4 L5 , it states that parameter values are samples from specified ranges assuming they are independent from each other. Also same thing on P8L17.

P6 L24: " ... on the GrIS evolution by testing..."

P6 L31: language: " This is also supported by estimate of both surface temperature anomalies" By and estimate??? by estimates of both surface temperature anomalies and... This part needs clarification.

P7 L7: "except for the fact that the oceanic changes associated with the millennial scale variability (deltaTmil,ocn) is set to zero"

P10 L31: small ocean temperature variations?

Figures:

Figure 2: This figure of the cube doesn't provide a clear visual of the distribution. I would like to see a something like figure 4 hear, but since the information is already in figure 4 the paper needs some modification. Figure 2 can be removed but there would have to be some major restructuring of the text in Sections 2.4 and 3.1.

Figure 3a: The black and blue colours are a poor choice as the lines are indiscernible. Contrasting colours would be much better or add more transparency to the lines.

Figure 7, SM2 and SM3: Same colour choice as in Figure 3.

---

## Author Comment (AC1) · 5 Mar 2019

The manuscript (MS) aims at estimating the Greenland contribution to sea level variability during the last glacial period in relation to D-O and Heinrich events. The authors apply an approach similar to that used in an already published paper concerned with glacial-interglacial cycles.

Overall, I think the exercise is very useful to give a first order estimate for the Greenland contribution to sea level variability during glacial times, and I think the approach of implementing ice-shelf ocean interaction is new and very relevant. I have one major point of concern, however, that I think needs consideration in order for the MS to make a good case.

It is very well established in the literature that during the last glacial period, the Heinrich events and the major ice rafting events in the North Atlantic are associated with the Greenland cold phases, the stadials. In contrast, during the mild Greenland periods, the interstadials, ice rafting and transport of continental material to the North Atlantic is much more limited. This is clearly illustrated in Figure 11 in the cited Hodell et al., QSR, 2010 paper, but also in many other studies where record resolution allows for a detailed comparison of marine records to the Greenland temperature record.

If I understand the model of the manuscript correctly, the ocean-terminating melt of GRiS is forced by the Greenland derived surface temperature, and the ocean temperature variability is assumed to be in phase with the (Greenland) atmosphere. Therefore, most of the marine-induced/basal melt from GRiS occurs during the interstadials. I see this as a highly unrealistic approach. Whereas the Greenland surface temperature is quite likely to give a good estimate for the Greenland surface melt during D-O events, I think this approach leads to a very unrealistic scenario for the basal melt that is mainly caused by interaction between the ice sheet and the subsurface ocean.

As seen in the MS figure 11, the modelled basal melt is completely out of phase with the melt events observed in the marine record. All the major modelled melt events occur during the Greenland interstadials, whereas all of the observed melt events occur during stadials. It is argued in the MS that the reason for this discrepancy could be that the source area for the marine IRD in this specific core could be different from Greenland. However, the timing of other IRD sources are also consistently occurring in the stadials. I think that the reason for this significant model-data disagreement is that the model approach of forcing the basal melt by Greenland surface temperatures is fundamentally wrong.

One could argue that is may not matter so much exactly when the ice sheet is losing or gaining mass as long as the inferred change in sea level variability is of the right order of magnitude. I this case, however, I think it is quite clear that the GRiS mass changes observed in the model are caused by an entirely wrong mechanism, and therefore are likely to be misleading and possibly of the wrong magnitude, also when it comes to sea level variability. Therefore, I think it is very important that a more realistic approach is applied for the basal melt/marine-terminating ice melt. It can possibly still be a simple approach, but it needs to include some estimated extent of sea ice in the North Atlantic, and some marine-based estimate of sub-surface ocean temperatures. The inclusion of sea ice is essential, because the sea ice partly seals off the ocean from direct heat exchange with the atmosphere, and thereby hampers the assumption of an in-phase temperature variability of ocean and atmospheric temperatures on the time scales relevant for D-O events.

Possibly, the authors can seek inspiration for a more realistic basal melt timing in those papers:

Dokken et al., Paleoceanography, 2013: Dansgaard-Oeschger cycles: Interactions between ocean and sea ice intrinsic to the Nordic seas.

Bassis et al., Nature, 2017: Heinrich events triggered by ocean forcing and modulated by isostatic adjustment.

We thank the reviewer for his constructive comments. Indeed, he raises a reasonable objection about the timing of the submarine melting peaks adopted in the manuscript, since most of the increased ice rafting events recorded into the North Atlantic during the last glacial period are indeed associated with Greenland stadials. This is supported by several sediment records coming from the Irminger Sea (Bond and Lotti, 1995; Van Kreveld et al., 2000; Rasmussen et al., 2016; Jonkers et al., 2010; Moros et al., 2002), the northern North Atlantic (Bond and Lotti, 1995; Barker et al., 2015; Hodell et al., 2010), the northwest flank of Iceland (Voelker and Haflidason, 2015) and the Nordic Seas (Rasmussen and Thomsen 2004). The Greenland ice sheet (GrIS) is proposed as one of the possible sources of the recorded ice rafted debris (IRDs) (Bond and

Lotti 1995; Moros et al. 2004; Prins et al., 2002), with a particular concentration of detritus coming from the East GrIS (Barker et al., 2015, Van Kreveld et al., 2000; Hodell et al., 2010; Andrews et al., 2017; Voelker and Haflidason, 2015; Rasmussen et al., 2016) and Northeast GrIS (Andrews et al., 2017), suggesting that the GrIS could have experienced intense ice mass variations throughout the Dansgaard-Oeschger (D-O) cycles.

The increase in iceberg discharge observed during cold stadials is attributed to warming of the subsurface, as many proxy records of the North Atlantic and Nordic Seas suggest (Ezat et al., 2014; Jonkers et al., 2010; Rasmussen and Thomsen 2004; Rasmussen et al., 2016; Sessford et al., 2018; Dokken et al., 2013). This decoupling between surface and subsurface during stadials is supported by modelling work as a result of reorganisations of North Atlantic Deep Water (NADW) formation (Vettoretti and Peltier, 2015; Brady and Otto-Bliesner, 2011; Mignot et al., 2007; Knutti et al., 2004; Marcott et al., 2011), and allows to explain the increase in ice rafting observed during stadials despite the low surface oceanic temperatures: warm subsurface waters would act as a trigger, or amplifier, of massive iceberg calving during the coldest stadials, such as the Heinrich events (Alvarez-Solas et al., 2011; Alvarez-Solas et al., 2013; Flückiger et al., 2006; Marcott et al., 2011; Bassis et al., 2017), and during the cold phases of D-Os (Shaffer et al., 2004; Petersen et al., 2013; Rasmussen et al., 2016; Boers et al., 2018).

The choice of a submarine melting signal in phase with atmospheric variations, i.e. surface warming during interstadials and surface cooling during stadials, was a simple approach based on the assumption that the Greenland continental shelf is relatively shallow. However, ice shelves are usually hundreds of meters thick and ocean-driven retreat processes are probably triggered by subsurface rather than surface waters. In the revised manuscript, the submarine melting rate evolution is now assumed to represent the conditions of oceanic waters at the subsurface, which are thought to be in antiphase with respect to the atmosphere (gradual stadial warming followed by interstadial cooling). For that, the interstadial-stadial oceanic temperature anomaly ($\Delta T^{mil}_{ocn}$) is now chosen to be negative (stadial oceanic temperature is higher than that during the interstadial), such that $\beta * \Delta T^{mil}_{ocn}$ (and the submarine melting rate) mirrors the subsurface peaks of warming during stadials. The timing of the basal melting signal is therefore similar to that of Alvarez-Solas et al., 2010, Alvarez-Solas et al., 2013, Bassis et al., 2017 and Boers et al., 2018. By doing so, now we no longer simulate the ice discharges during Greenland Interstadials but, during stadials in a very consistent manner when compared with proxies (as pointed out by the referee; please see current Figure 8).

Many authors associate D-Os to changes in sea-ice cover over the Nordic Seas (Hoff et al., 2016; Dokken 2013; Li et al., 2010; Sime et al., 2019; Jensen et al., 2018). The absence of synchronicity between atmospheric and ocean warming due to the insulating effect of sea ice in the Nordic Seas during the stadials is now implicitly taken into account by our forcing by considering increased submarine melting during stadials. Perturbing the model with spatially variable subsurface oceanic temperatures, which may reflect variations in sea-ice cover, instead of a simple temperature time series, would likely affect the simulation of local ice mass variations throughout the last glacial cycle. Moreover, the few available model reconstructions of stadial-interstadial oceanic temperatures suggest that subsurface warming increases almost homogeneously across the ocean during the stadials, at least in the Nordic Seas (Fig S2 of Zhang et al., 2014, and Fig. 2 of Alvarez-Solas et al., 2018). This is also in agreement with large-scale subsurface warming suggested by Rasmussen and Thomsen, 2004 and Marcott et al. 2011. Thus, considering a spatial variation in oceanic temperatures (and sea ice) around the GrIS may not be fundamental for this study, which primarily aims to look at the response of the whole GrIS to D-Os, and the simplified approach of perturbing the ice-sheet model with a spatially constant oceanic forcing should be sufficient for our purposes.

All these points have been discussed in the new version of the manuscript.

Specifically, the following paragraph has been added in Section 2.3:

"*$\Delta T^{orb}_{ocn}$ and $\Delta T^{mil}_{ocn}$ are the glacial-interglacial and interstadial-stadial oceanic temperature anomalies (K), respectively. $(1 - \alpha(t)) * \Delta T^{orb}_{ocn}$ reflects the long timescales variations resulting from orbital changes. $\beta * \Delta T^{mil}_{ocn}$ expresses the millennial-scale temperature changes at the subsurface assumed to be in antiphase with respect to the Greenland atmospheric temperature inferred from Greenland ice cores (e.g. Johnsen et al., 2001; Kindler et al., 2014). Thus, we are assuming that subsurface water temperatures increase during stadials and decrease during interstadials. This is in agreement with the presence of warming subsurface waters during stadial periods as suggested by several records of the North Atlantic and Nordic Seas (Ezat et al., 2014; Jonkers et al., 2010; Rasmussen and Thomsen 2004; Rasmussen et al., 2016; Sessford et al., 2018; Dokken et al., 2013) and supported by modelling work (Vettoretti and Peltier, 2015; Brady and Otto-Bliesner, 2011; Mignot et al., 2007; Knutti et al., 2004; Marcott et al., 2011). The result is a submarine melting signal that peaks during D-O stadials. This is in line with the temporal evolution of oceanic forcings used to inspect the effect of subsurface warming during the coldest stadials, i.e. Heinrich events, by*

*perturbing other ice-sheet models (Alvarez-Solas et al., 2010; Alvarez-Solas et al., 2013; Bassis et al., 2017), or, as done recently, to investigate the origin of D-O events through a conceptual model (Boers et al., 2018).*"

Also, the following paragraph has been added in the Discussion:

"*Many authors associate D-O events to changes in sea-ice cover over the Nordic Seas (Dokken et al., 2013; Hoff et al., 2016; Jensen et al., 2018; Li et al., 2010; Sime et al., 2019), however our ice-sheet model does not resolve sea-ice processes. Nevertheless, the absence of synchronicity between atmospheric and ocean warming due to the insulating effect of sea ice in the Nordic Seas during the stadials is implicitly taken into account by our forcing which associates peaks in submarine melting to stadials. We are aware that spatially-variable fields should be taken into account for an exhaustive investigation of the problem. Perturbing the model with spatially variable subsurface oceanic temperatures, which may reflect variations in sea-ice cover, instead of a simple temperature timeserie, would likely affect the simulation of local ice mass variations throughout the last glacial cycle. Both stadial-interstadial and glacial-interglacial temperature anomalies could be taken from existing transient model outputs, for instance. However, a complete map of the observed PD (present-day) basal melting rates for the whole Greenland domain does not exist yet, in contrast to Antarctica (Rignot et al., 2013), thus limiting the effectiveness of including additional complexity at this time. The fact that the few available model reconstructions of stadial-interstadial oceanic temperatures suggest that subsurface warming increases almost homogeneously across the ocean during the stadials, at least in the Nordic Seas (Zhang et al., 2014b; Alvarez-Solas et al., 2018), supports the choice of ignoring spatial oceanic variations as a first approach. This is also in agreement with large-scale subsurface warming suggested by Rasmussen and Thomsen (2004) and Marcott et al. (2011). Thus, considering a spatial variation in oceanic temperatures (and sea ice) around the GrIS may not be fundamental for this study, which primarily aims to look at the response of the whole GrIS to D-O events, and the simplified approach of perturbing the ice-sheet model with a spatially constant oceanic forcing should be sufficient for our purposes.*"

Also, changing the oceanic forcing in the ice-sheet model has had implications in the analysis of the results and in the description of the experimental design. A new version of the manuscript tracking the changes will make the improvements to the manuscript clear.

REFERENCES:

Alvarez-Solas et al., 2010. Links between ocean temperature and iceberg discharge during Heinrich events. *Nature Geoscience*, 3, 2, 122.

Alvarez-Solas et al., 2011. Heinrich event 1: an example of dynamical ice-sheet reaction to oceanic changes. *Climate of the Past*, 7, 4, 1297-1306.

Alvarez-Solas et al., 2013. Iceberg discharges of the last glacial period driven by oceanic circulation changes. *PNAS*, 110, 41, 16350-16354.

Alvarez-Solas et al., 2018. Oceanic forcing of the Eurasian Ice Sheet on millennial time scales during the Last Glacial Period, *Clim. Past Discuss.*, https://doi.org/10.5194/cp-2018-89, in review.

Andrews et al., 2017. Denmark Strait during the late glacial maximum and marine isotope stage 3: Sediment sources and transport processes. *Marine Geology*, 390, 181-198.

Barker et al., 2015. Icebergs not the trigger for North Atlantic cold events. *Nature*, 520, 7547, 333.

Bassis et al., 2017. Heinrich events triggered by ocean forcing and modulated by isostatic adjustment. *Nature*, 542, 7641, 332.

Boers et al., 2018. Ocean circulation, ice shelf, and sea ice interactions explain Dansgaard-Oeschger cycles. *PNAS*, 115, 47, E11005-E11014.

Bond and Lotti, 1995. Iceberg discharges into the North Atlantic on millennial time scales during the last glaciation. *Science*, 267(5200), 1005-1010.

Brady and Otto-Bliesner, 2011. The role of meltwater-induced subsurface ocean warming in regulating the Atlantic meridional overturning in glacial climate simulations. *Climate dynamics*, 37, 7-8, 1517-1532.

Dokken et al., 2013. Dansgaard-Oeschger cycles: Interactions between ocean and sea ice intrinsic to the Nordic seas. *Paleoceanography and Paleoclimatology*, 28, 3, 491-502.

Ezat et al., 2014. Persistent intermediate water warming during cold stadials in the southeastern Nordic seas during the past 65 ky. *Geology*, 42, 8, 663-666.

Flückiger et al., 2006. Oceanic processes as potential trigger and amplifying mechanisms for Heinrich events. *Paleoceanography*, 21, 2.

Hodell et al., 2010. Phase relationships of North Atlantic ice-rafted debris and surface-deep climate proxies during the last glacial period. *Quaternary Science Reviews*, 29, 27-28, 3875-3886.

Hoff et al., 2016. Sea ice and millennial-scale climate variability in the Nordic seas 90 kyr ago to present. *Nature communications*, 7, 12247.

Jensen et al., 2018. A spatiotemporal reconstruction of sea-surface temperatures in the North Atlantic during Dansgaard-Oeschger events 5-8, *Clim. Past*, 14, 901-922.

Jonkers et al., 2010. A reconstruction of sea surface warming in the northern North Atlantic during MIS 3 ice-rafting events. *Quaternary Science Reviews*, 29, 15-16, 1791-1800.

Knutti et al., 2004. Strong hemispheric coupling of glacial climate through freshwater discharge and ocean circulation. *Nature*, 430, 7002, 851.

van Kreveld et al., 2000. Potential links between surging ice sheets, circulation changes, and the Dansgaard-Oeschger cycles in the Irminger Sea, 60–18 kyr. *Paleoceanography and Paleoclimatology*, 15, 4, 425-442.

Li et al., 2010. Can North Atlantic sea ice anomalies account for Dansgaard–Oeschger climate signals?. *Journal of climate*, 23, 20, 5457-5475.

Marcott et al., 2011. Ice-shelf collapse from subsurface warming as a trigger for Heinrich events. *PNAS,* 108, 33, 13415-13419.

Mignot et al., 2007. Atlantic subsurface temperatures: Response to a shutdown of the overturning circulation and consequences for its recovery. *Journal of Climate*, 20, 19, 4884-4898.

Moros et al., 2002. Were glacial iceberg surges in the North Atlantic triggered by climatic warming?. *Marine Geology*, 192, 4, 393-417.

Moros et al., 2004. Sea surface temperatures and ice rafting in the Holocene North Atlantic: climate influences on northern Europe and Greenland. *Quaternary Science Reviews*, 23, 20-22, 2113-2126.

Petersen et al., 2013. A new mechanism for Dansgaard-Oeschger cycles. *Paleoceanography*, 28, 1, 24-30.

Prins et al., 2002. Ocean circulation and iceberg discharge in the glacial North Atlantic: Inferences from unmixing of sediment size distributions. *Geology*, 30, 6, 555-558.

Rasmussen et al., 2016. North Atlantic warming during Dansgaard-Oeschger events synchronous with Antarctic warming and out-of-phase with Greenland climate. *Scientific reports*, 6, 20535.

Rasmussen and Thomsen, 2004. The role of the North Atlantic Drift in the millennial timescale glacial climate fluctuations. *Palaeogeography, Palaeoclimatology, Palaeoecology*, 210, 1, 101-116.

Sessford et al., 2018. High-Resolution Benthic Mg/Ca Temperature Record of the Intermediate Water in the Denmark Strait Across D-O Stadial-Interstadial Cycles. *Paleoceanography and Paleoclimatology*, 2018, 33, 11, 1169-1185.

Shaffer et al., 2004. Ocean subsurface warming as a mechanism for coupling Dansgaard-Oeschger climate cycles and ice-rafting events. *Geophysical Research Letters*, 31, 24.

Sime et al., 2019. Impact of abrupt sea ice loss on Greenland water isotopes during the last glacial period. *PNAS*, 201807261.

de Vernal et al., 2013. Sea ice in the paleoclimate system: the challenge of reconstructing sea ice from proxies - an introduction. *Quaternary Science Reviews*, *79*, 1-8.

Vettoretti and Peltier, 2015. Interhemispheric air temperature phase relationships in the nonlinear Dansgaard-Oeschger oscillation. *Geophysical Research Letters*, 42, 4, 1180-1189.

Voelker and Haflidason, 2015. Refining the Icelandic tephrachronology of the last glacial period–the deep-sea core PS2644 record from the southern Greenland Sea. *Global and Planetary Change*, 131, 35-62.

//Zhang et al., 2014. Instability of the Atlantic overturning circulation during Marine Isotope Stage 3. *Geophysical Research Letters*, 41, 12, 4285-4293.

---

## Author Comment (AC2) · 5 Mar 2019

Tabone et al. use a three dimensional hybrid ice-sheet-shelf model forced by oceanic fluctuations derived from paleo records. The work studies the role of millennial scale climate variability in ocean-ice sheet interaction and is a topic that is of great interest to the glacial climate variability. These type of studies provide a basis to test our earth system models under past climate conditions in order to validate projections of anthropogenic climate change or to develop a better process understanding of climate components that are critical for assessing future anthropogenic climate change. The introduction provides a nice review of the literature on the subject and reviews the key research questions on ocean-ice sheet interaction. The major problem problem I see with the ocean-ice sheet coupling is that the oceanic forcing is not realistic and the study would be better suited to investigate the sensitivity of the model to marine shelf instability as a result of sub-surface temperature variations. Page 11, Line 22 states that the model seems to lag Heinrich Events by 2-3 ka. The manuscript goes to great length to compare the model with sediment cores when the forcing (Figure 1c) is representative of surface temperature changes. The authors state that (Page 12, L2) that they do not aim to precisely reconstruct the timing and spatial distribution of ice discharge during D-O events. I don't think it is satisfactory to show that oceanic millennial scale variability is influencing the GrIS evolution alone unless something can be said about a process based explanation of what is happening in the real climate system based on the modelling results.

We are grateful to the reviewer for their valuable comments. Indeed, the objection regarding the usage of an oceanic forcing that follows surface instead of subsurface temperature variations is legitimate. Since ice shelves are usually hundreds of meters thick, their instability is likely more affected by subsurface rather than surface waters. This assumption is also valid for the effect that the ocean may have on grounding-line fluctuations, since grounding zones are typically found several hundreds of meters deep in the ocean (at least at the present day, e.g. Wilson et al., 2017). We therefore agree that perturbing the model through a surface-dependent oceanic forcing might have been a weakness of the experimental design. Using surface or subsurface oceanic temperatures has a strong impact when investigating the role of the ocean during the last glacial period. Evidence points to a strong surface-subsurface temperature decoupling during the Dansgaard-Oeschger (DO) cycles (e.g. Vettoretti and Peltier, 2015; Brady and Otto-Bliesner, 2011; Knutti et al., 2004), suggesting that subsurface waters have experienced warming during stadials and cooling during interstadials (e.g. Ezat et al., 2014; Sessford et al., 2018; Dokken et al., 2013). To account for this decoupling and to consider subsurface oceanic temperatures instead, we now force the ice-sheet model through an oceanic temperature anomaly signal assumed to be in antiphase with respect to that of the oceanic surface, which in turn reflects atmospheric changes (following e.g. Alvarez-Solas et al. (2010), Alvarez-Solas et al. (2013), Bassis et al. (2017) and Boers et al. (2018)).

Although this modification did not change the overall conclusions of the work and it mainly affected the quantitative description of the results, the manuscript has been deeply changed in this respect. For example, current Figure 8 shows now the comparison between our simulated ice flux and IRDs. The good agreement between the two indicates that GrIS likely contributed to the observed iceberg discharges of the Northern Hemisphere during the last glacial period (LGP), and that these were triggered through oceanic millennial-scale forcing. We believe this aspect together with the rest of our conclusions configure "a process based explanation of what is happening in the real climate system based on the modelling results". We recommend checking the new version of the manuscript for a complete view of the improvements.

Answers to specific comments are reported below.

Other comments:

Section 2.1 Page 4, Lines 15-31: The description of coarse grid points is not clear here. In line 30, the ice sheet model is given (20x20km) but the ocean resolution forcing is not described clearly.

The oceanic forcing is considered as spatially homogeneous, i.e. simply given by the time evolution of the submarine melting rate as specified by Eq. 6. In that equation, since both orbital and millennial-scale temperature anomalies ($\Delta T^{orb}_{ocn}$ and $\Delta T^{mil}_{ocn}$) and the reference basal melting rate $B_{ref}$ are considered as spatially constant, the resulting basal melting rate $B(t)$ is spatially homogeneous too. Therefore, the spatial resolution of the ocean is irrelevant.

To make this clear we added the sentence in Section 2.4:

"*These variables are here all considered as spatially uniform around Greenland for the sake of simplicity, leading to a spatially homogeneous basal melting rate.* "

Section 2.1 and 2.2: P4, L31-32; P5 L2; P13 L13 The statement "we consider the atmosphere as modulated by orbital changes". This statement should be rewritten to downplay the impression that there is an atmosphere in the model. Statements such as L31-32 that describe millennial variability in the atmosphere are also misleading. One mainly thinks of atmospheric dynamics in terms of atmospheric climate/weather variability which happens on short timescales. Trace gases, insolation and other slowly evolving atmospheric properties are the result of the internal and external forcing of the earth system.

We agree with the referee and have modified the text accordingly:

The sentence of P4, L31-32 has been rewritten. It now appears as: "*Since the goal of this work is to investigate the sensitivity of the GrIS to past millennial-scale variability in the ocean, we force the ice-sheet model through spatially variable surface atmospheric temperatures and precipitation that only reflect orbital variations.*"

The sentence of P5, L2 has been rewritten as: "*The atmospheric forcing only includes the orbital-scale evolution of temperature and precipitation over the last glacial cycle, thus changes associated with short (millennial) timescales are not taken into account here.*"

The sentence of P13, L13 has been rewritten as: "*Finally, atmospheric precipitation and temperature used to perturb the ice-sheet model vary only at long (orbital) timescales, while shorter (millennial-scale) variations related to the DO events are not considered here.* "

Section 2.3 P5 L26: "... changes in ocean temperature into..."

This sentence has been changed accordingly.

P6 L4-7: The construction of beta is a nice measure of millennial scale variability. But if you really want to influence the basal melt rate during the glacial as proposed by some of the studies in the introduction in a realistic fashion the millennial index in Figure 1c should be inverted to reflect changes in subsurface temperature during Heinrich and D-O stadials. This is a major problem with the study. There is some of this discussion in Section 4.2 on Model limitations and caveats but this detracts from realism of the science and what the study can actually say about what processes are important for millennial scale variability.

As already pointed out in the general comments, we agree with the reviewer that considering subsurface rather than surface oceanic temperatures might be more accurate when considering the effect of oceanic variation on ice-shelf destabilisation and grounding-line retreat. Since oceanic variations at the subsurface associated to DO events are assumed to vary in antiphase with respect to those at the surface (as suggested by several proxies and model results), it is reasonable to simply invert the submarine melting rate signal in the way that subsurface warmings correspond to stadials and subsurface coolings to interstadials. This is possible for example by considering the interstadial-stadial oceanic temperature anomaly $\Delta T^{mil}_{ocn}$ as negative (warmer waters during stadials). In this way, and leaving the millennial-scale climatic index $\beta$ as it was in the previous version of the manuscript, the submarine melting rate is now of the form that warming peaks appear during stadials.

The experimental design, part of the analysis of the results and discussion have been changed accordingly.

P6 L7-8: What is meant by the statement that both deltaTorb,ocn and deltaTmil,ocn are both assumed to be in phase with the atmosphere when there is no deltaTmil,atm in equation (1) and (2).

We agree that the sentence may be misleading. Since now $\Delta T^{mil}_{ocn}$ reflects changes in subsurface waters, the sentence has been modified to: "*$\Delta T^{orb}_{ocn}$ and $\Delta T^{mil}_{ocn}$ are the glacial-interglacial and interstadial-stadial oceanic temperature anomalies (K), respectively. $(1 - \alpha(t)) * \Delta T^{orb}_{ocn}$ reflects the long timescales variations resulting from orbital changes. $\beta * \Delta T^{mil}_{ocn}$ expresses the millennial-scale temperature changes at the subsurface assumed to be in antiphase with respect to the Greenland atmospheric temperature inferred from Greenland ice cores (e.g. Johnsen et al., 2001; Kindler et al., 2014). Thus, we are assuming that subsurface water temperatures increase during stadials and decrease during interstadials. This is in agreement with the presence of warming subsurface waters during stadial periods as suggested by several records of the North Atlantic and Nordic Seas (Ezat et al., 2014; Jonkers et al., 2010; Rasmussen and Thomsen 2004; Rasmussen et al., 2016; Sessford et al., 2018; Dokken et al., 2013) and supported by modelling work (Vettoretti and Peltier, 2015; Brady and Otto-Bliesner, 2011; Mignot et al., 2007; Knutti et al., 2004; Marcott*

*et al., 2011). The result is a submarine melting signal that peaks during DO stadials. This is in line with the temporal evolution of oceanic forcings used to inspect the effect of subsurface warming during the coldest stadials, i.e. Heinrich events, by perturbing other ice-sheet models (Alvarez-Solas et al., 2010; Alvarez-Solas et al., 2013; Bassis et al., 2017), or, as done recently, to investigate the origin of DO events through a conceptual model (Boers et al., 2018).*"

P6 L12: Adding a short description of how changes in RSL on the orbital timescale are prescribed might be more helpful instead of just the reference.

This sentence has been changed to: "*Changes in global sea level at orbital timescales are prescribed in the model and are used to compute the grounding-line position, which is based on a simple flotation criterion. The applied sea-level variation time series is taken from Bintanja and Van de Wal (2008). The signal is inferred from a marine benthic oxygen isotope record reconstructed for the last 3 Myr through an ice-sheet model coupled to a simple marine temperature model.*"

Section 2.4 P6 L15-20: The authors state that the basal melt is dependant on 4 parameters. Another problem I see is in the variation of the parameter changes during the LHS sampling. The reference basal melt is given as Bref =kappa(Tclim,ocn - T_f) where T_f is fixed. Equation (6) has B proportional to kappa*deltaTorb,ocn. So variations in kappa and Bref are not independent. Changes in kappa will make inverse changes in Tclim,ocn (the mean climatology of the ocean) if Bref is varied in an inconsistent manner. So am I missing some understanding of the variational procedure or is the LHS sampling (which considers previous choices) taking care of this discrepancy? Again in section 2.4 L5 , it states that parameter values are samples from specified ranges assuming they are independent from each other. Also same thing on P8L17.

The reviewer makes a good point here. Probably simply stating that the four key parameters ($B_{ref}$, $\kappa$, $\Delta T^{orb}_{ocn}$ and $\Delta T^{mil}_{ocn}$) of the basal melting rate equation (Eq. 6) are independent from each other is not entirely accurate, since, as the reviewer says, $B_{ref}$ does depend on $\kappa$. However, $B_{ref}$ also depends on $T_{clim,ocn}$ and $T_f$. $T_{clim,\,ocn}$ is the climatological mean of the water temperature and $T_f$ is the freezing point temperature, both at the grounding line. These two terms are largely unconstrained since they may be at least depth-dependent (Beckmann and Goosse, 2003). Defining $B_{ref}$ in the equation is, therefore, a simplification made to elude the hard choice of assigning values to these two variables. If $T_{clim,ocn} - T_f$ is set as constant, $B_{ref}$ is directly proportional to $\kappa$ (as we did in another work, Tabone et al., 2018b, where $T_{clim,ocn} - T_f$ was set to 1K). However, if this quantity is left free to vary (implicitly), $B_{ref}$ can be considered as *conditionally* independent from $\kappa$, where the condition is considering $T_{clim,ocn} - T_f$ as an additional degree of freedom. The choice of decoupling $B_{ref}$ from $\kappa$ follows the aim of investigating as many cases as possible to have a good sample of simulations to work with. Nevertheless, the investigated values of $B_{ref}$ (0-10 m a$^{-1}$) and $\kappa$ (0-10 m a$^{-1}$ K$^{-1}$) are chosen to be consistent with plausible values of $T_{clim,ocn} - T_f$ (the median of all analysed values ~ 0.9 K), while those that are less probable ($T_{clim,ocn} - T_f > 2$ K) are discarded from the analysis a posteriori since they show a very low millennial-scale variability (high $B_{ref}$ and very low $\kappa$). Thus, in a certain way we do take care to avoid the discrepancy pointed out by the reviewer, although not through the LHS sampling technique.

To clarify this concept, Section 2.4 has been rewritten as:

"*Following the discussion above, we can rewrite the basal melting equation (Eq. 4) as:*

$$B(t) = B_{ref} + \kappa \left( (1 - \alpha(t)) \, \Delta T^{orb}_{ocn} + \beta \, \Delta T^{mil}_{ocn} \right)$$

*Written in this form, the basal melting formulation depends on the choice of four parameter values: $B_{ref}$, $\kappa$, $\Delta T^{orb}_{ocn}$ and $\Delta T^{mil}_{ocn}$. These variables are here all considered as spatially uniform around Greenland for the sake of simplicity, leading to a spatially homogeneous basal melting rate. To assess the GrIS response to millennial-scale variability in the ocean we could simply consider varying the value of $\kappa$, which is the sensitivity of the oceanic forcing (Tabone et al., 2018a). However, by construction of Eq. 6, increasing $\kappa$ does not necessarily mean increasing the millennial-scale oceanic effect alone, since this would enhance concurrently both the millennial and the orbital-scale components in the ocean. Therefore, investigating the oceanic millennial-scale variability effect on the past GrIS is not as straightforward as expected. Moreover, none of the four parameters of Eq. 6 is perfectly constrained in reality, and a sensitivity study on the influence of their chosen values on the GrIS evolution would be required. For these reasons, it is first useful to characterise the impact of millennial-scale variability in the ocean on the GrIS by testing a broad range of values of the key-parameters in Eq. 6.*

*Some considerations need to be made before describing the experiments. First, it should be noted that $B_{ref}$ depends on $\kappa$ (Eq. 4), thus any change in $\kappa$ results in a change in $B_{ref}$ too. However, $B_{ref}$ also depends on the water and freezing-point temperatures at the grounding zone ($T_{clim,ocn}$ and $T_f$, respectively), that are largely*

*unconstrained, since they mostly depend on the characteristics of the considered grounding line, e.g. on the depth (Beckmann and Goosse, 2003), and in principle can be represented by a broad range of values. We could have set $T_{clim,ocn} - T_f$ to a constant value to make $B_{ref}$ scale directly with $\kappa$, but this would limit the range of submarine melting rates to be investigated and the possibility of better constraining these two terms, which are still poorly known around Greenland. On the contrary, here, $T_{clim,ocn} - T_f$ is treated as an additional degree of freedom, that, although not explicitly perturbed in the equation, allows us to consider $B_{ref}$ as independent from $\kappa$. Of course, decoupling these two variables requires additional attention in considering a range of values for $B_{ref}$ and $\kappa$ that allow for realistic $T_{clim,ocn} - T_f$. The simulations that do not fulfill this requirement will be discarded a posteriori in the analysis.*

*Second, by construction, the $\alpha$ and $\beta$ indices share the same normalisation. Thus the glacial-interglacial and the interstadial-stadial subsurface oceanic temperature anomalies have the same amplitudes. This is also supported by estimates of subsurface temperatures at both short (millennial) and long (orbital) timescales. Reconstructed LGM-present day subsurface anomalies, which at orbital timescales are considered to follow those at the surface, are between 0 and -3 K around Greenland (Annan and Hargreaves, 2013; MARGO 2009). A similar range of values is found for the interstadial-stadial subsurface temperature anomalies (Alvarez-Solas et al., 2018; Brady and Otto-Bliesner, 2011; Vettoretti and Peltier, 2015; Zhang et al., 2014). The problem is therefore reduced to three degrees of freedom: $\Delta T^{mil}_{ocn}$, set to vary between 0 and 3 K, $B_{ref}$, between 0 and 10 m a$^{-1}$ (chosen as a reasonable climatic mean between Rignot et al. (2010), Rignot et al. (2016), Straneo et al. (2012) and Wilson et al. (2017) for the largest tidewater glaciers around the GrIS, and Liu et al. (2015) and Rignot et al. (2013) for Antarctica) and $\kappa$, between 0 and 10 m a$^{-1}$ K$^{-1}$ (following Rignot and Jacobs (2002) for Antarctica).*

*To test a wide range of combinations between the three key parameters, we perform a large ensemble (LE) of model simulations using the near-random Latin Hypercube Sampling (LHS) technique (McKay et al., 1979), which allows us to efficiently explore the phase-space of the key parameters minimising the LE computational cost with respect to the full-factorial sampling technique. The LHS method has already been used to constrain different ice-sheet model parameters and to assess their influence on the model's behavior (Applegate et al., 2012; Stone et al., 2010; Stone et al., 2013; Robinson et al., 2017). The parameter values are sampled from the specified ranges and, assuming, as discussed, that they are independent from each other, they are randomly combined to generate a total LE of 100 simulations, named TOT simulations. At the same time, we perform another set of identical simulations, except for the fact that the climatic index associated with the millennial scale variability ($\beta$) is set to zero. These are named ORB simulations and are used for direct comparison with the TOT simulations, as discussed in Section 3. See Table 1 for a full list of the phase-space of parameters investigated within the two LEs. To initialise the model we use the present-day topography of Greenland from Schaffer et al. (2016). All the simulations of the LE cover the last two glacial cycles, with the first considered as a spin up and therefore not analysed. "*

P6 L24: " ... on the GrIS evolution by testing..."

Sentence changed accordingly (see Section 2.4).

P6 L31: language: " This is also supported by estimate of both surface temperature anomalies" By and estimate??? by estimates of both surface temperature anomalies and... This part needs clarification.

This sentence has been changed since the forcing method has been modified (now we consider subsurface instead of surface oceanic temperatures). See Section 2.4.

P7 L7: "except for the fact that the oceanic changes associated with the millennial scale variability (deltaTmil,ocn) is set to zero"

This sentence has been changed to: "*At the same time, we perform another set of identical simulations, except for the fact that the climatic index associated with the millennial scale variability ($\beta$) is set to zero*". See Section 2.4.

P10 L31: small ocean temperature variations?

Sentence changed accordingly.

Figures:

Figure 2: This figure of the cube doesn't provide a clear visual of the distribution. I would like to see a something like figure 4 here, but since the information is already in figure 4 the paper needs some

modification. Figure 2 can be removed but there would have to be some major restructuring of the text in Sections 2.4 and 3.1.

Figure 2 has been substituted by one table (Table 1), that reviews the parameter values investigated in the two LEs. Since the information of Figure 2 is still reported, we don't think there is the need of reorganizing the text much.

Figure 3a: The black and blue colours are a poor choice as the lines are indiscernible. Contrasting colours would be much better or add more transparency to the lines.

Blue lines of this figure (now Figure 2a) are now changed to red lines.

Figure 7, SM2 and SM3: Same colour choice as in Figure 3.

Blue lines of Figure SM2 and SM3 are now changed to red lines.

REFERENCES:

Alvarez-Solas et al., 2010. Links between ocean temperature and iceberg discharge during Heinrich events. *Nature Geoscience*, 3, 2, 122.

Alvarez-Solas et al., 2013. Iceberg discharges of the last glacial period driven by oceanic circulation changes. *PNAS*, 110, 41, 16350-16354.

Alvarez-Solas et al., 2018. Oceanic forcing of the Eurasian Ice Sheet on millennial time scales during the Last Glacial Period, *Clim. Past Discuss.*, https://doi.org/10.5194/cp-2018-89, in review.

Annan and Hargreaves, 2013. A new global reconstruction of temperature changes at the Last Glacial Maximum. *Climate of the Past,* 9, no 1, p. 367-376.

Applegate et al., 2012. An assessment of key model parametric uncertainties in projections of Greenland Ice Sheet behavior. *The Cryosphere*, 6, no 3, p. 589-606.

Bassis et al., 2017. Heinrich events triggered by ocean forcing and modulated by isostatic adjustment. *Nature*, 542, 7641, 332.

Beckmann and Goosse, 2003. A parameterization of ice shelf–ocean interaction for climate models. *Ocean modelling,* 5, no 2, p. 157-170.

Bintanja and Van de Wal, 2008. North American ice-sheet dynamics and the onset of 100,000-year glacial cycles. *Nature*, 454, no 7206, p. 869.

Brady and Otto-Bliesner, 2011. The role of meltwater-induced subsurface ocean warming in regulating the Atlantic meridional overturning in glacial climate simulations. *Climate dynamics*, 37, 7-8, 1517-1532.

Dokken et al., 2013. Dansgaard-Oeschger cycles: Interactions between ocean and sea ice intrinsic to the Nordic seas. *Paleoceanography and Paleoclimatology*, 28, 3, 491-502.

Ezat et al., 2014. Persistent intermediate water warming during cold stadials in the southeastern Nordic seas during the past 65 ky. *Geology,* 42, 8, 663-666.

Johnsen et al., 2001. Oxygen isotope and palaeotemperature records from six Greenland ice-core stations: Camp Century, Dye-3, GRIP, GISP2, Renland and NGRIP. *Journal of Quaternary Science,* 16, 4, 299-307.

Jonkers et al., 2010. A reconstruction of sea surface warming in the northern North Atlantic during MIS 3 ice-rafting events. *Quaternary Science Reviews*, 29, 15-16, 1791-1800.

Kindler et al., 2014. Temperature reconstruction from 10 to 120 kyr b2k from the NGRIP ice core. *Climate of the Past,* 10, 2, p. 887-902.

Knutti et al., 2004. Strong hemispheric coupling of glacial climate through freshwater discharge and ocean circulation. *Nature*, 430, 7002, 851.

Liu et al., 2015. Ocean-driven thinning enhances iceberg calving and retreat of Antarctic ice shelves. *PNAS*, 112, no 11, p. 3263-3268.

Marcott et al., 2011. Ice-shelf collapse from subsurface warming as a trigger for Heinrich events. *PNAS*, 108, 33, 13415-13419.

MARGO project members 2009. Constraints on the magnitude and patterns of ocean cooling at the Last Glacial Maximum. *Nature Geoscience*, 2, no 2, p. 127.

McKay et al., 1979. Comparison of three methods for selecting values of input variables in the analysis of output from a computer code, *Technometrics*, 21, 239-245.

Mignot et al., 2007. Atlantic subsurface temperatures: Response to a shutdown of the overturning circulation and consequences for its recovery. *Journal of Climate*, 20, 19, 4884-4898.

Rasmussen et al., 2016. North Atlantic warming during Dansgaard-Oeschger events synchronous with Antarctic warming and out-of-phase with Greenland climate. *Scientific reports*, 6, 20535.

Rasmussen and Thomsen 2004. The role of the North Atlantic Drift in the millennial timescale glacial climate fluctuations. *Palaeogeography, Palaeoclimatology, Palaeoecology*, 210, 1, 101-116.

Rignot et al., 2010. Rapid submarine melting of the calving faces of West Greenland glaciers. *Nature Geoscience*, 3, 3, p. 187.

Rignot et al., 2013. Ice-shelf melting around Antarctica. *Science*, vol. 341, no 6143, p. 266-270.

Rignot et al., 2016. Modeling of ocean-induced ice melt rates of five west Greenland glaciers over the past two decades. *Geophysical Research Letters*, 43, no 12, p. 6374-6382.

Rignot and Jacobs, 2002. Rapid bottom melting widespread near Antarctic ice sheet grounding lines. *Science*, 296, no 5575, p. 2020-2023.

Robinson et al., 2017. MIS-11 duration key to disappearance of the Greenland ice sheet. *Nature communications*, 8, p. 16008.

Schaffer et al., 2016. A global, high-resolution data set of ice sheet topography, cavity geometry, and ocean bathymetry. *Earth Syst. Sci. Data*, 8, 543–557.

Sessford et al., 2018. High-Resolution Benthic Mg/Ca Temperature Record of the Intermediate Water in the Denmark Strait Across D-O Stadial-Interstadial Cycles. *Paleoceanography and Paleoclimatology*, 33, 11, 1169-1185.

Stone et al., 2010. Investigating the sensitivity of numerical model simulations of the modern state of the Greenland ice-sheet and its future response to climate change. *The Cryosphere*, 4, no 3, p. 397-417.

Stone et al., 2013. Quantification of the Greenland ice sheet contribution to Last Interglacial sea level rise. *Climate of the Past*, 9, no 2, p. 621-639.

Straneo et al., 2012. Characteristics of ocean waters reaching Greenland's glaciers. *Annals of Glaciology*, 53, no 60, p. 202-210.

Tabone et al., 2018a. The sensitivity of the Greenland Ice Sheet to glacial-interglacial oceanic forcing. *Climate of the Past,* 14, 455-472.

Tabone et al., 2018b. Submarine melt as a potential trigger of the NEGIS margin retreat during MIS-3. *The Cryosphere Discussion*, https://doi.org/10.5194/tc-2018-228, in review.

Vettoretti and Peltier, 2015. Interhemispheric air temperature phase relationships in the nonlinear Dansgaard-Oeschger oscillation. *Geophysical Research Letters*, 42, 4, 1180-1189.

Wilson et al., 2017. Satellite-derived submarine melt rates and mass balance (2011–2015) for Greenland's largest remaining ice tongues. *The Cryosphere*, 11, 2773–2782.

Zhang et al., 2014. Instability of the Atlantic overturning circulation during Marine Isotope Stage 3. *Geophysical Research Letters*, 41, 12, 4285-4293.

---

## Author Response (AR2)

**Answer to Editor Decision: Publish subject to minor revisions (review by editor)**
(16 Mar 2019) by Marit-Solveig Seidenkrantz

Comments to the Author:
Dear Dr. Tabone,

Thanks you for re-submitting your manuscript for consideration to "Climate of the Past". I am pleased to see your detailed answers to the reviewers' comments, as well as thorough corrections to your manuscript, in particular with respect to surface vs subsurface ocean temperatures during stadials vs interstadials of the DO cycles.

However, before final decision, I would ask you for a few additional corrections in order to make your results and conclusions clearer for the reader:
1) Please add both in the abstract and the conclusions that you use warm surface waters (and cool subsurface waters?) during D-O interstadials and cold surface waters and warmer subsurface waters during stadials to drive your model simulation.
2) Please also make it clear during which periods (D-O stadials or interstadials) that you see ice surface vs basal melting. This may be logical, but it would still be good to point out. The conclusions and abstract also lack a clear conclusion on, whether the total melting rates are higher due to surface or basal melt according to your model output. In the conclusions, you could also point out the iceberg release that may be a consequence of increased basal melt.
There may be parts where you do not feel comfortable in making a very clear statement, but in that case a tentative conclusion would be good.

Again, when re-submitting your manuscript, please include a version, where all changes are highlighted clearly.

Kind regards,
Marit-Solveig Seidenkrantz, Editor

Dear Editor,
thank you very much for your suggestions.

The manuscript has been modified accordingly by adding a couple of sentences in both abstract and conclusions.

Specifically, the abstract has been changed as:

*"Temperature reconstructions from Greenland ice sheet (GrIS) ice cores indicate the occurrence of more than twenty abrupt warmings during the Last Glacial Period (LGP) known as Dansgaard-Oeschger (D-O) events. Although their ultimate cause is still debated, evidence from both proxy data and modelling studies robustly links these to reorganisations of the Atlantic Meridional Overturning Circulation (AMOC). During the LGP, the GrIS expanded as far as the continental shelf break and was thus more directly exposed to oceanic changes than in the present. Therefore oceanic temperature fluctuations on millennial timescales could have had a non-negligible impact on the GrIS. Here we assess the effect of millennial-scale oceanic variability on the GrIS evolution from the last interglacial to the present day. To do so, we use a three-dimensional hybrid ice-sheet-shelf model forced by subsurface oceanic temperature fluctuations, assumed to increase during D-O stadials and decrease during D-O interstadials. Since in our model the atmospheric forcing follows orbital variations only, the increase in total melting at millennial timescales is a direct result of an increase in basal melting. We show that the GrIS evolution during the LGP could have been strongly influenced by oceanic changes on millennial timescales, leading to ocean-induced ice volume contributions above 1 m SLE. Also, our results suggest that the increased flux of GrIS icebergs as inferred from North Atlantic proxy records could have been triggered, or intensified, by peaks in melting at the base of the ice shelves resulting from increasing subsurface oceanic temperatures during D-O stadials. Several regions across the GrIS could thus have been responsible for ice mass discharge during D-O events, opening the possibility of a non-negligible role of the GrIS in oceanic reorganisations throughout the LGP."*

Also, the conclusions have been modified as:

*"We have assessed the effect of the millennial-scale oceanic variability on the evolution of the GrIS during the LGP. To do so, we used an ice-sheet-shelf model, in which the millennial-scale variability in the ocean was imposed as a fluctuation in the basal melting rate at the grounding line and below the ice shelves, \*resulting from oceanic temperature anomalies at the subsurface, where warmer (cooler) waters are*

*associated with D-O stadials (interstadials). We first characterised the millennial variability through a sensitivity test for a broad range of values of the perturbed parameters in the submarine melting equation. We showed that the millennial-scale contribution to ice-volume variations during the LGP could have reached peaks of more than 1 m SLE. The southeastern area around Kangerdlugssuaq fjord, Baffin Bay and the NEGIS regions were found to be very sensitive to millennial-scale variability in the ocean. Ice thicknesses simulated at the marine margin differed by 500-1000 m from those simulated by orbital-only driven oceanic variations. Moreover, imprints of these differences are still found for several tens (hundreds - in certain regions) of kilometers far from the ice-ocean interface due to the velocity-driven upstream propagation of the ice-flow perturbation. Although the aim of this work was far from assessing the true timing and spatial distribution of any GrIS ice discharge that occurred during the D-O events, we showed that considering the millennial-scale variability in the ocean is necessary to reproduce some of the IRD peaks observed in proxy data. Specifically, the basal melting increase during D-O stadials, associated with warmer oceanic waters at the subsurface, could have been responsible for the enhanced release of Greenland icebergs as inferred from North Atlantic sediment cores. Our work thus suggests that millennial-scale induced changes in ocean circulation and temperature may be important drivers of the GrIS evolution during the LGP, advancing the hypothesis of a potential role of the GrIS in oceanic reorganisations at millennial timescales."*

We hope the message of the paper is clearer now.

Best regards,

Ilaria Tabone and co-authors

[revised manuscript text omitted]